# k-Maximum Inner Product Attention for Graph Transformers and the Expressive Power of GraphGPS

**Jonas De Schouwer**
Stanford University
jonasds@cs.stanford.edu

**Haitz Sáez de Ocáriz Borde**
University of Oxford
chri6704@ox.ac.uk

**Xiaowen Dong**
University of Oxford
xiaowen.dong@eng.ox.ac.uk

## Abstract

Graph transformers have shown promise in overcoming limitations of traditional graph neural networks, such as oversquashing and difficulties in modelling long-range dependencies. However, their application to large-scale graphs is hindered by the quadratic memory and computational complexity of the all-to-all attention mechanism. Although alternatives such as linearized attention and restricted attention patterns have been proposed, these often degrade performance or limit expressive power. To better balance efficiency and effectiveness, we introduce k-Maximum Inner Product (k-MIP) attention for graph transformers. k-MIP attention selects the most relevant key nodes per query via a top-k operation, yielding a sparse yet flexible attention pattern. Combined with an attention score computation based on symbolic matrices, this results in linear memory complexity and practical speedups of up to an order of magnitude compared to all-to-all attention, enabling the processing of graphs with over 500k nodes on a single A100 GPU. We provide a theoretical analysis of expressive power, showing that k-MIP attention does not compromise the expressiveness of graph transformers: specifically, we prove that k-MIP transformers can approximate any full-attention transformer to arbitrary precision. In addition, we analyze the expressive power of the GraphGPS framework, in which we integrate our attention mechanism, and establish an upper bound on its graph distinguishing capability in terms of the S-SEG-WL test. Finally, we validate our approach on the Long Range Graph Benchmark, the City-Networks benchmark, and two custom large-scale inductive point cloud datasets, consistently ranking among the top-performing scalable graph transformers.

## 1 Introduction

Since the advent of the original Graph Neural Network (GNN) model (Scarselli et al., 2009), graph machine learning has become a powerful tool for analyzing relational data across domains ranging from social networks (Borisyuk et al., 2024; Fan et al., 2019) to molecular biology (Jumper et al., 2021; Fout et al., 2017) and recommendation systems (He et al., 2020; Wang et al., 2019), and has become a cornerstone of Geometric Deep Learning (Bronstein et al., 2017; 2021).

Traditional approaches typically follow the message-passing paradigm, where information is propagated locally between neighbouring nodes. Despite its effectiveness, this paradigm has certain shortcomings such as oversmoothing (Li et al., 2018), oversquashing (Alon & Yahav, 2020) and limited expressive power (Morris et al., 2019; Xu et al., 2018; Loukas, 2019), all of which may lead to difficulties in capturing long-range dependencies (Akansha, 2023). Graph transformers have recently emerged as a promising alternative (Ying et al., 2021; Dwivedi & Bresson, 2020; Rampášek et al., 2022), enabling information exchange between every pair of nodes. However, this comes at the cost of quadratic complexity, and it remains an open question how to adapt the Transformer

architecture (Vaswani et al., 2017) to scale effectively to large graphs with potentially millions of nodes. In this paper we propose the k-Maximum Inner Product (k-MIP) self-attention mechanism for graph transformers. k-MIP attention dynamically selects the $k$ most influential keys for each query based on the intermediate activations, achieving scalability while avoiding the drawbacks of linearisation, rigid attention patterns and graph subsampling, which we discuss in Section 2.2.

The main contributions of this paper are the following.

- We introduce k-Maximum Inner Product (k-MIP) self-attention for graph transformers, which achieves linear memory complexity and yields up to a ten-fold speedup over full attention[1]. As a result, k-MIP attention enables the processing of graphs with over 500k nodes on a single A100 GPU.

- We show that k-MIP attention can be seamlessly integrated into the GraphGPS framework and provide a theoretical analysis of its expressive power, establishing an upper bound on the graph-distinguishing capability of GraphGPS in terms of the S-SEG-WL test (Zhu et al., 2023). This analysis clarifies how positional and structural encodings enable expressivity in graph transformers.

- We prove that k-MIP transformers can approximate any full-attention transformer to arbitrary precision, thereby guaranteeing that the proposed sparsification does not reduce the expressive power of transformer-based architectures.

- We empirically demonstrate competitive performance against other scalable graph transformers on a range of benchmarks, including the Long Range Graph Benchmark (LRGB) (Dwivedi et al., 2022), the City-Networks benchmark (Liang et al., 2025), and two custom large-scale inductive point cloud datasets based on ShapeNet-Part (Yi et al., 2016) and S3DIS (Armeni et al., 2016).

## 2   RELATED WORK

This section provides background on the k-MIP attention mechanism and surveys related work on scalable graph transformers. An extended discussion can be found in Appendix G.

### 2.1   k-MAXIMUM INNER PRODUCT ATTENTION

The attention mechanism used in this paper was first introduced by Zhao et al. (2019) in the context of natural language processing, revisited by Gupta et al. (2021), and later applied to computer vision by Wang et al. (2022). While these works demonstrated the mechanism's effectiveness in their respective domains, their methods suffered from quadratic memory complexity, rendering them unsuitable for direct adaptation to large graph datasets. To handle the substantially larger scale of the benchmarks considered in this work, we enhance the k-MIP attention mechanism through the use of symbolic matrices (Charlier et al., 2021), similar to previous work in latent graph inference (Kazi et al., 2023; Borde et al., 2023b;a) (see Appendix F for more details). This optimization achieves linear memory complexity and accelerates computation by an order of magnitude compared to full attention. Although the computational complexity remains quadratic, it nevertheless enables processing of graphs with over 500k nodes, as in City-Networks (Liang et al., 2025), on a single A100 GPU. Furthermore, we provide a novel theoretical exploration of its expressive power in section 4. We also note that other approximate variants of k-MIP attention have recently been proposed in the literature (Zeng et al., 2025; Mao et al., 2024).

### 2.2   GRAPH TRANSFORMERS

Inspired by the success of Transformers (Vaswani et al., 2017) in natural language modeling, substantial effort has been devoted to adapting this architecture to the graph domain (Borde, 2024). Most graph transformers achieve this by treating graph nodes as tokens, since using edges and/or subgraphs as tokens would lead to a combinatorial explosion in the number of tokens for large graphs. Early graph transformers, such as Graphormer (Ying et al., 2021) and Graph Transformer (Dwivedi & Bresson, 2020), demonstrated strong performance. However, they inherited the quadratic memory complexity of the Transformer, restricting their applicability to graphs with at most a few thousand nodes. To overcome this scalability hurdle, various approaches have been proposed since then,

---

[1]In our implementation; performance depends on the KeOps backend and GPU memory regime.

which can be broadly classified into four categories with examples provided in table 1. (1) **Graph subsampling** approaches process only sampled portions of large graphs but are consequently unable to capture long-range dependencies. (2) Methods with **engineered attention patterns** restrict attention along predefined patterns, potentially missing out on important relationships between nodes. (3) **Linearized attention** methods approximate standard attention through kernel tricks or matrix factorization, achieving linear complexity but often at the cost of performance. (4) Methods with **learnable attention patterns** dynamically determine which node pairs are relevant for attention in order to focus computational resources on the most important connections. These approaches promise both efficiency gains and strong predictive performance, but remain relatively underexplored in the literature.

Having established this categorization, we position k-MIP attention within the *learnable patterns* category. A comparison to other methods is provided in Appendix G.1. Note that some frameworks, notably GraphGPS (Rampášek et al., 2022), are designed to be modular with respect to the attention mechanism and can thus belong to (or integrate) any of the above categories. Also, while some prior works (Shehzad et al., 2024) classify attention-based MPNNs, such as GAT (Velickovic et al., 2017), as graph transformers, we exclude these methods from our analysis.

Table 1: Categorization of scalable graph transformer methods.

| Graph Subsampling | Engineered Patterns | Linearized Attention | Learnt Patterns |
|---|---|---|---|
| Gophormer (Zhao et al., 2021) NAGphormer (Chen et al., 2022b) | Exphormer (Shirzad et al., 2023) GPS+BigBird (Rampášek et al., 2022) SpExphormer (Shirzad et al., 2024) | Nodeformer (Wu et al., 2022) Difformer (Wu et al., 2023) SGFormer (Wu et al., 2024) GPS+Performer (Rampášek et al., 2022) | GOAT (Kong et al., 2023) **GPS+k-MIP (ours)** |

## 3 METHOD

In standard multi-head self-attention, every query attends to every key to compute dense attention score matrices $\boldsymbol{A}^h \in [0,1]^{N \times N}$, where $h$ denotes the attention head. While this enables global information exchange between any pair of nodes, it comes with two significant drawbacks when applied to large graphs. First, the quadratic time and memory complexity in $N$ makes full attention computationally infeasible for graphs with more than a few thousand nodes. This limitation severely restricts the applicability of graph transformers to real-world problems, where graphs often have millions of nodes or more. Second, most attention scores tend to be small, as each node typically has only a few truly relevant connections. This observation is supported in Appendix I.1. This sparsity suggests that most attention computations are effectively wasted on irrelevant node pairs and may introduce noise into the learning process. Our goal is to address both issues simultaneously: the attention mechanism should be computationally efficient while focusing only on the most relevant node interactions. Since the attention scores provide a natural measure of relevance between nodes (Vaswani et al., 2017), we propose to only attend to the $k$ keys with the highest attention score for each query. We use a fixed $k$ across all queries to enable efficient representation of the top-$k$ operation results. We study the influence of this parameter in Appendix I.2.

### 3.1 k-MAXIMUM INNER PRODUCT SELF-ATTENTION

We propose k-Maximum Inner Product self-attention, which extends multi-head self-attention by restricting each query to attend only to the $k$ keys with the highest inner product scores. Formally, given an input matrix $\boldsymbol{X} \in \mathbb{R}^{N \times d}$, a number of attention heads $H$ and learnable weight matrices $\boldsymbol{W}_Q^h, \boldsymbol{W}_K^h \in \mathbb{R}^{d \times d_K}$ and $\boldsymbol{W}_V^h \in \mathbb{R}^{d \times d_V}$, k-MIP attention operates as follows.

$$\boldsymbol{A}^h = \text{softmax}\left(\frac{1}{\sqrt{d_K}} \mathcal{T}_k\left(\boldsymbol{X}\boldsymbol{W}_Q^h(\boldsymbol{X}\boldsymbol{W}_K^h)^\top\right)\right), \quad h \in \{1, \ldots, H\}. \tag{1}$$

$$\text{k-MIP}(\boldsymbol{X}) = \sum_{h=1}^{H} \boldsymbol{A}^h \boldsymbol{X}\boldsymbol{W}_V^h \boldsymbol{W}_O^h. \tag{2}$$

where $\mathcal{T}_k$ is the top-k operation that retains only the $k$ largest elements in each row and sets all other elements to $-\infty$. See the pseudocode in Appendix E. For efficient implementation, we store the intermediate matrix $\boldsymbol{X}\boldsymbol{W}_Q^h(\boldsymbol{X}\boldsymbol{W}_K^h)^\top$ as a symbolic matrix (Feydy et al., 2020), which represents

the matrix as a formula rather than materializing all elements in memory (Appendix F). Only when applying the top-k reduction are the $N^2$ elements computed lazily in the GPU's registers, avoiding memory overflows and costly transfers to the GPU's global memory. While we could not avoid the quadratic computational complexity of computing each inner product due to the hardness of searching in high-dimensional spaces, this approach gives the algorithm a linear memory footprint and a speedup of an order of magnitude compared to full attention, as we confirm in section 5.1. Furthermore, the top-k indices can be reused in the sparse backpropagation computation, which makes the **backward pass virtually negligible** in computation for large numbers of tokens.

## 4 EXPRESSIVE POWER

The expressive power of a parametrized model refers to the range of functions it can represent. Understanding the expressiveness of a model is crucial for identifying its limitations and for designing models that can solve a given task. For feedforward neural networks, research on expressivity led to the *universal approximation theorem*, which states that a network with a single hidden layer and sufficiently many neurons can approximate any continuous function on a compact subset of $\mathbb{R}^n$ under mild conditions on the activation function (Cybenko, 1989; Funahashi, 1989; Hornik, 1991; Leshno et al., 1993). Similarly, a more recent paper (Yun et al., 2019) has shown that full-attention transformer networks of constant width and sufficient depth can approximate any continuous sequence-to-sequence function to arbitrary precision.

For models that learn from graph-structured data, the question of expressive power is considerably more intricate, as a model's ability to incorporate node features, graph structure, edge weights, and edge features all contribute to its overall expressiveness. Although there are numerous ways to quantify the expressive power of such methods, important upper bounds have been derived for MPNNs by studying their ability to distinguish non-isomorphic graphs. In particular, it has been shown that traditional MPNNs are at most as powerful as the Weisfeiler–Lehman (1-WL) test for detecting graph isomorphisms (Xu et al., 2018; Morris et al., 2019). As a consequence, MPNNs are unable to represent any function that assigns different values to graphs that the 1-WL test cannot distinguish.

We extend this line of work by establishing a comparable upper bound on the expressive power of the GraphGPS framework using the SEG-WL test from Zhu et al. (2023). A key advantage of grounding our analysis in the SEG-WL framework is that it positions our results within an established hierarchy of expressiveness results: Zhu et al. (2023) have already characterized several other graph transformer architectures (Dwivedi & Bresson, 2020; Ying et al., 2021; Kreuzer et al., 2021; Zhao et al., 2021; Chen et al., 2022a) in terms of SEG-WL, so our result directly enables comparison of the expressive power of GraphGPS (and hence GPS+k-MIP) to all of these methods on a common footing. We provide additional relevant background on this topic in Appendix A, including a more extensive literature review and mathematical preliminaries for our results, where we dive into node coloring and a description of the SEG-WL test.

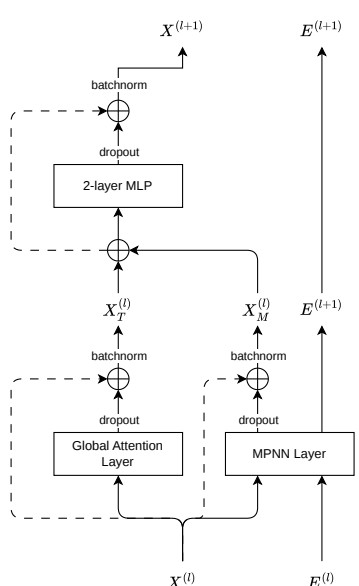

Figure 1: One layer in the GraphGPS framework. The dashed lines indicate residual connections. Based on Rampášek et al. (2022).

### 4.1 EXPRESSIVE POWER OF GRAPHGPS

GraphGPS is a framework for building graph transformers, introduced in (Rampášek et al., 2022) and later leveraged by Exphormer (Shirzad et al., 2023) and GPS++ (Masters et al., 2022), among others. The framework consists of many sequentially stacked layers of the type depicted in Figure 1. The $l$-th layer updates both node and edge feature matrices, transforming $(\boldsymbol{X}^{(l-1)}, \boldsymbol{E}^{(l-1)})$ into $(\boldsymbol{X}^{(l)}, \boldsymbol{E}^{(l)})$. Each layer consists of three main components: the *MPNN Layer*, the *Global Attention Layer*, and the *2-layer MLP*. The MPNN Layer is a traditional message-passing neural network layer. In this work, we employ

GatedGCN for all experiments, as it was observed in the GraphGPS paper to yield the best performance (Rampášek et al., 2022). While the MPNN Layer is limited to message passing along the given graph, the Global Attention Layer ensures that nodes that are not directly connected can attend to each other. The Global Attention Layers examined in this work include Transformer (i.e., full attention) (Vaswani et al., 2017), BigBird (Zaheer et al., 2020), Performer (Choromanski et al., 2020), Exphormer (Shirzad et al., 2023), and k-MIP attention. With the exception of Exphormer, each of these allows *any two nodes* to attend to each other. The 2-layer MLP is a simple two-layer feedforward neural network that is applied to the node after the MPNN and Global Attention Layers.

In this section, we prove that the GraphGPS framework (which includes GPS+k-MIP, Exphormer Shirzad et al. (2023), and GPS++ Masters et al. (2022)) can only distinguish graphs that are also distinguishable by the SEG-WL test (see Appendix A.2) enhanced with the same positional and structural encodings. This result allows us to compare the expressive power of GraphGPS to the 1-WL test and highlights the importance of expressive encodings. Further, we will use it to shed new perspective on the super-1-WL expressiveness results given in various previous works (Rampášek et al., 2022; Kreuzer et al., 2021; Shirzad et al., 2023). Importantly, it should be noted that this result is not necessarily a drawback to our method, as we will show in section 4.2 that k-MIP transformers can universally approximate any function representable by a full-attention transformer.

First, we prove that the graph distinguishing power of the GraphGPS framework is upper bounded by the $S$-SEG-WL test (see Definition 7 in Appendix A.2) that uses the same node and edge feature enhancements.

**Theorem 1.** *Let $\mathcal{A}$ be an instance of the GraphGPS framework that enhances its node features with $\nu(\boldsymbol{A})$ and its edge features with $\mu(\boldsymbol{A})$. Then $\mathcal{A}$ can only distinguish graphs that are also distinguishable by the $S$-SEG-WL test, where $S = (f_A, f_R)$ and $f_A, f_R$ are defined as*

$$f_A(v, G) = \nu(\boldsymbol{A})_v, \tag{3}$$

$$f_R(v, u, G) = \begin{cases} (0, \mathbb{1}_{u=v}, \boldsymbol{0}_{d_{edge}}, \boldsymbol{0}_{d_{PE}}) & \text{if } (u,v) \notin E \\ (1, \mathbb{1}_{u=v}, \boldsymbol{E}_{uv}, \mu(\boldsymbol{A})_{uv}) & \text{if } (u,v) \in E \end{cases}, \tag{4}$$

*where the set of possible colors is $\mathcal{C} = \mathbb{R}^{d_{PE}} \cup \{0,1\}^2 \times \mathbb{R}^{d_{edge}} \times \mathbb{R}^{d_{PE}} \cup \mathbb{R}^d \cup \mathbb{R}^{d_{edge}}$.*

The proof of this theorem can be found in Appendix B.1. By establishing a connection between the expressive power of GraphGPS and the $S$-SEG-WL test, we embed the GraphGPS framework (and hence, our GPS+k-MIP) into the same expressiveness hierarchy that Zhu et al. (2023) constructed for other graph transformers, making it possible to directly compare GraphGPS to these methods as well as to the 1-WL test. We will discuss three implications of this result. First, we compare the expressive power of GraphGPS to the 1-WL test. Second, we investigate the effect of more expressive encodings on the expressive power of GraphGPS. Third, we discuss the origin of the expressive power of graph transformers.

**Comparison to the 1-WL test**     The $S$-SEG-WL test generalizes the 1-WL test: 1-WL is recovered by choosing, for all $u, v \in V$,

$$f_A(v, G) = c_0, \tag{5}$$

$$f_R(v, u, G) = \begin{cases} c_1 & \text{if } (u,v) \in E \\ c_2 & \text{if } (u,v) \notin E \end{cases}, \tag{6}$$

where $c_0, c_1, c_2 \in \mathcal{C}$ are constants. Consequently, in the standard setting for evaluating 1-WL expressiveness (identical node features, identical edge features, and no positional encodings) note that the structural encoding scheme $S$ from Theorem 1 degenerates to the form eq. (5)-eq. (6). Hence, Theorem 1 implies that the expressive power of the GraphGPS framework in terms of graph distinguishability is upper bounded by the 1-WL test.

**When more expressive encodings are used**     As we formally introduce in Appendix A.2.3, there is a preorder on the expressive powers of $S$-SEG-WL test. In particular, if $S, S'$ are structural encoding schemes and $S'$ is a refinement of $S$, then $S'$-SEG-WL is at least as expressive in terms of graph distinguishability as $S$-SEG-WL. While this theorem does not state that $S'$-SEG-WL is strictly more expressive than $S$-SEG-WL, Zhu et al. (2023) provide various examples where the order relation is

strict. The implication for GraphGPS is that more expressive positional and structural encodings lead to a higher upper bound on the expressiveness of the framework. In particular, using the Laplacian positional encoding allows GraphGPS to distinguish some graphs that are indistinguishable by the 1-WL test. An example of such a pair is given in Kreuzer et al. (2021). Combining this with the fact that the MPNN module in the GraphGPS framework can be equally expressive as the 1-WL test when an expressive MPNN is used (Xu et al., 2018), we can conclude that GraphGPS with Laplacian positional encodings is strictly more expressive than the 1-WL test.

**The origin of graph transformers' expressive power**   While various previous works on graph transformers (Rampášek et al., 2022; Shirzad et al., 2023; Kreuzer et al., 2021) have claimed super-1-WL expressiveness for their methods (in terms of graph distinguishing power), our result highlights that this expressiveness comes from the positional and structural encodings applied to the node and edge features, rather than from the Transformer architecture itself. In particular, for all of the three mentioned works, the same level of graph distinguishing power could be attained using an expressive GNN where the input features are augmented with the same positional and structural encodings.

## 4.2   Universal Approximation of Full-Attention Transformers

Since k-MIP attention is a sparsification of the standard multi-head self-attention, it is natural to ask what range of functions it can represent. To address this question, we prove that for each full-attention transformer $T_{full}$ there exists a k-MIP transformer $T_{\text{k-MIP}}$ that approximates $T_{full}$ to an arbitrary approximation error $\epsilon > 0$ on any compact set $U \subseteq \mathbb{R}^{N \times d}$. First, we present a unified definition for both full-attention and k-MIP transformers. Both are compositions of transformer blocks, parameterized by their attention mechanism. The input and output tokens are the rows of the respective matrices. This complements the results from Yun et al. (2020), as discussed in Appendix D.4.

**Definition 1** (General transformer block). *A transformer block $t_{\mathcal{A}}^{h,m,r}$ is a row-permutation equivariant mapping from $\mathbb{R}^{N \times d}$ to $\mathbb{R}^{N \times d}$ that implements the following sequential operation[2]:*

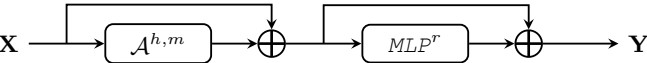

*Here, $\mathcal{A}^{h,m}$ is an attention layer (e.g., full attention or k-MIP attention) with $h$ heads and hidden dimension $d_K = d_V = m$. $MLP^r$ is a 2-layer MLP with ReLU activation and hidden dimension $r$. Note that here we ignore normalization for simplicity.*

**Definition 2** (Class $\mathcal{T}_{\mathcal{A}}^{h,m,r}$). *$\mathcal{T}_{\mathcal{A}}^{h,m,r}$ is the class of transformers using attention mechanism $\mathcal{A}$, where each transformer is a composition of an arbitrary number of transformer blocks $t_{\mathcal{A}}^{h,m,r}$:*

$$\mathcal{T}_{\mathcal{A}}^{h,m,r} := \left\{ T_{\mathcal{A}} : \mathbb{R}^{N \times d} \to \mathbb{R}^{N \times d} \mid T_{\mathcal{A}} \text{ is a composition of transformer blocks } t_{\mathcal{A}}^{h,m,r} \right\} \quad (7)$$

Regarding the two cases considered in this work, $\mathcal{T}_{full}^{h,m,r}$ is the class of transformers using full multi-head self-attention and $\mathcal{T}_{\text{k-MIP}}^{h,m,r}$ is the class of transformers using k-MIP attention.

**Theorem 2** (k-MIP Approximation Theorem). *Consider any full-attention transformer $T_{full} \in \mathcal{T}_{full}^{h,m,r}$, any $\epsilon > 0$, any $p \in [1, \infty)$, and any compact $U \subseteq \mathbb{R}^{N \times d}$. Then there exists a k-MIP transformer $T_{\text{k-MIP}} \in \mathcal{T}_{\text{k-MIP}}^{2,1,4}$ such that*

$$\left( \int_U ||T_{\text{k-MIP}}(\boldsymbol{X}) - T_{full}(\boldsymbol{X})||_p^p d\boldsymbol{X} \right)^{1/p} < \epsilon. \quad (8)$$

This theorem is proven in Appendix C and discussed in Appendix D, including detailed comparisons to prior theoretical work, scope, and limitations.

While Theorem 2 shows that k-MIP *transformers* can approximate any full-attention transformer, note that it is not necessarily true that the constituent k-MIP *transformer blocks* will approximate

---

[2]We assume a deterministic, permutation-equivariant tie-breaking rule for the top-k operator; this is standard in theoretical analyses and does not affect practical behavior.

every corresponding full-attention transformer block. Indeed, in Appendix I.1 we show that this is not the case: a k-MIP attention layer is a very poor approximation of the full attention layer with the same weights until $k$ approaches $N$, because the top-$k$ keys by inner product capture only a small fraction of the total attention weight. Yet, Theorem 2 proves that the composition of many such layers has the expressive power to approximate any full-attention transformer to arbitrary precision.

## 5 EXPERIMENTS

We evaluate the efficiency, prediction quality, and scalability enabled by the k-MIP self-attention mechanism through three sets of experiments: (1) computational efficiency in a controlled setting, (2) prediction quality on LRGB (Dwivedi et al., 2022), and (3) scalability to large-scale graph datasets (Liang et al., 2025; Chang et al., 2015; Yi et al., 2016). For in-depth experimental details refer to Appendix H. We provide complementary performance measurements in Appendix J.

### 5.1 OBJECTIVE 1: COMPUTATIONAL EFFICIENCY

We compare runtime and memory usage of different attention mechanisms while varying the number of nodes $N \in \{10^{i/2} \mid i = 4, \ldots, 12\}$ with $d_K = 10$ and $k = 10$. We consider both an *inference setting* (forward pass only, no gradient tracking) and a *training setting* (forward and backward pass, with gradient tracking). Results are displayed in fig. 2 and a breakdown of the computational cost is provided in fig. 3. For experimental details see Appendix H.2.

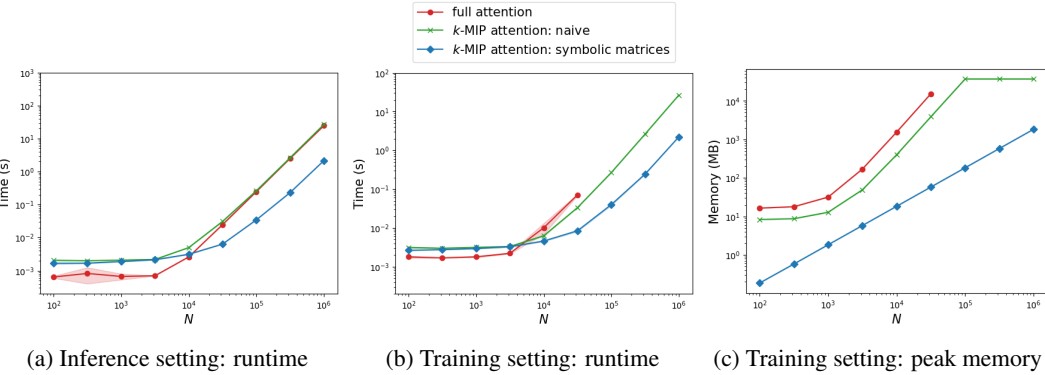

(a) Inference setting: runtime     (b) Training setting: runtime     (c) Training setting: peak memory

Figure 2: Comparison of full attention and k-MIP attention (with/without symbolic matrices). Shown is the mean $\pm$ std over 5 runs, measured on a single 40GB A100 GPU. Full attention gave OOM errors in the training setting for $N \geq 10^5$. The data behind this figure can be found in Appendix J.

**Results**    k-MIP attention achieves a speedup of an order of magnitude over full attention in our implementation. Specifically, k-MIP is $12.43\times$ faster during inference at $N = 10^6$ tokens and $8.46\times$ faster during training at $N = 10^{4.5}$ tokens. Crucially, while full attention encounters OOM errors for $N \geq 10^5$, k-MIP scales to $N = 10^7$ nodes on a single 40GB A100 GPU thanks to its linear memory complexity. A per-stage profiling breakdown (fig. 3) reveals that the top-k search dominates the forward pass of k-MIP attention, while the remaining stages are negligible by comparison. Notably, the backward pass is nearly instantaneous, since the top-k indices do not require recomputation during backpropagation (see Appendix E). For a direct comparison with FlashAttention (Dao et al., 2022) refer to Appendix K.

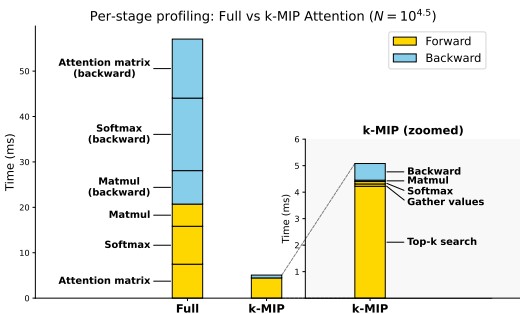

Figure 3: Runtime breakdown of full attention and k-MIP attention in the training setting at $N = 10^{4.5}$, measured on a single 40GB A100 GPU. The left and middle bars compare full and k-MIP attention at the same scale; the right panel zooms into k-MIP attention.

Table 2: Test performance of GPS+k-MIP and baselines on LRGB. Shown is the mean $\pm$ std over four runs. The **first**, second, and third best results are highlighted. [†]Taken from (Tönshoff et al., 2023). [‡]Taken from (Shirzad et al., 2023).

| | PascalVOC-SP (F1↑) | COCO-SP (F1↑) | Pept-Func (AP↑) | Pept-Struct (MAE↓) |
|---|---|---|---|---|
| **GCN**[†] | $20.78 \pm 0.31$ | $13.38 \pm 0.07$ | $68.60 \pm 0.50$ | $0.2460 \pm 0.0007$ |
| **GINE**[†] | $27.18 \pm 0.54$ | $21.25 \pm 0.09$ | $66.21 \pm 0.67$ | $0.2473 \pm 0.0017$ |
| **GAT** | $27.15 \pm 0.49$ | $18.86 \pm 0.11$ | $67.87 \pm 0.56$ | $0.2488 \pm 0.0018$ |
| **GatedGCN**[†] | $38.80 \pm 0.40$ | $29.22 \pm 0.18$ | $67.65 \pm 0.47$ | $0.2477 \pm 0.0009$ |
| **GPS + BigBird** | $38.75 \pm 0.64$ | $35.25 \pm 0.27$ | $64.83 \pm 0.73$ | $0.2566 \pm 0.0019$ |
| **GPS + Performer** | $37.92 \pm 1.13$ | $27.70 \pm 0.27$ | $66.06 \pm 0.64$ | $0.2643 \pm 0.0008$ |
| **GPS + Transformer**[†] | $44.40 \pm 0.65$ | $38.84 \pm 0.55$ | $65.34 \pm 0.91$ | $0.2509 \pm 0.0014$ |
| **Exphormer**[‡] | $39.75 \pm 0.37$ | $34.55 \pm 0.09$ | $65.27 \pm 0.43$ | $0.2481 \pm 0.0007$ |
| **GPS + k-MIP (ours)** | $39.69 \pm 0.92$ | $35.56 \pm 0.45$ | $66.27 \pm 0.44$ | $0.2562 \pm 0.0037$ |

## 5.2 OBJECTIVE 2: PREDICTION QUALITY

We aim to assess the empirical performance of k-MIP attention in graph transformers compared to other scalable attention mechanisms. To this end, we integrate five attention mechanisms into the GraphGPS architecture (see section 4.1) and compare them against each other and against MPNN baselines on LRGB (Dwivedi et al., 2022). We follow the methodology of Tönshoff et al. (2023) with a 500k parameter budget (details in Appendix H.3).

**Results** Table 2 presents our evaluation on LRGB. The results show that MPNNs outperform the GTs on both Peptides datasets, in line with previous work (Tönshoff et al., 2023). On PascalVOC-SP and COCO-SP, by contrast, graph transformers excel. On both of those, k-MIP attention matches or outperforms other scalable attention mechanisms, though the gap with full attention remains large.

## 5.3 OBJECTIVE 3: SCALABILITY

We now evaluate k-MIP attention's performance on large graphs using two benchmarks: (1) City-Networks (Liang et al., 2025) for transductive learning on road network graphs, and (2) point cloud segmentation datasets ShapeNet-Part (Yi et al., 2016) and S3DIS (Armeni et al., 2016) converted to k-NN graphs for inductive learning. Experimental details are provided in Appendix H.4 and Appendix H.5, respectively.

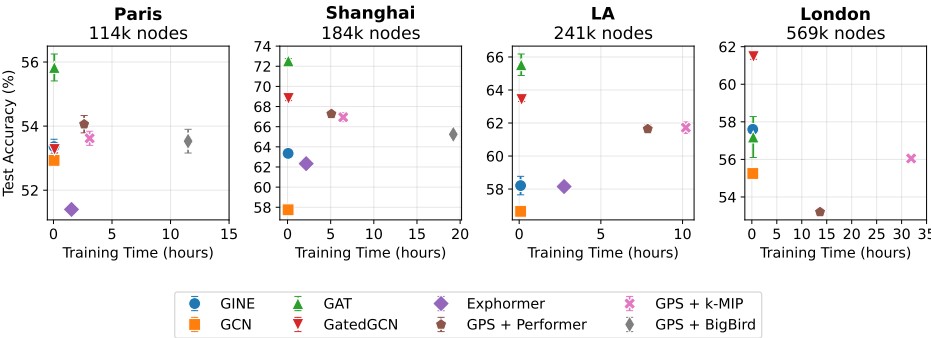

Figure 4: Tradeoffs between training time and accuracy across different datasets in City-Networks (Liang et al., 2025). Shown is the mean $\pm$ std over four runs, except for the London dataset where we only ran one run for the graph transformer models. GPS+BigBird was not evaluated on LA due to long training times. GPS+Transformer, Exphormer, and GPS+BigBird returned OOM on London.

**Results on City-Networks** Figure 4 presents the results on the City-Networks benchmark. The performance comparison reveals that GAT outperforms all graph transformer variants. Among graph transformers, GPS+k-MIP achieves comparable accuracy to GPS+Performer and significantly outperforms Exphormer and GPS+BigBird. Crucially, GPS+k-MIP scales to the London dataset

Table 3: Test performance of GPS+k-MIP and baselines on ShapeNet-Part and S3DIS. Shown is the mean $\pm$ std over three runs. The **first**, second, and third best results are highlighted.

|  | ShapeNet-Part (F1↑) | S3DIS (mIoU↑) |
|---|---|---|
| GCN | $60.18 \pm 0.04$ | $39.98 \pm 1.47$ |
| GINE | $64.57 \pm 0.35$ | $44.16 \pm 0.62$ |
| GAT | $63.01 \pm 0.17$ | $44.24 \pm 1.14$ |
| GatedGCN | $76.20 \pm 0.32$ | $63.71 \pm 1.28$ |
| GPS + BigBird | $79.65 \pm 0.98$ | $67.92 \pm 0.91$ |
| GPS + Performer | $77.36 \pm 1.23$ | $60.83 \pm 0.56$ |
| GPS + Transformer | – | – |
| Exphormer | $82.62 \pm 0.31$ | **$68.37 \pm 0.23$** |
| GPS + k-MIP (ours) | **$82.68 \pm 0.64$** | $67.99 \pm 1.51$ |

with 569k nodes on a single 80GB A100 GPU, showcasing its ability to handle real-world large graphs. This outcome could be attributed to the absence of positional encodings in the benchmark: computing Laplacian eigenvectors becomes computationally prohibitive at this scale, depriving graph transformers of the expressivity boost discussed in Section 4.1.

**Results on ShapeNet-Part and S3DIS**    Table 3 presents the results for the ShapeNet-Part and S3DIS benchmarks. Graph transformers clearly outperform MPNNs on both tasks. GPS+k-MIP achieves the best performance on ShapeNet-Part, roughly on par with Exphormer and outperforming GPS+Performer and GPS+BigBird by a significant margin. On S3DIS, GPS+k-MIP is only outperformed by Exphormer.

## 6    CONCLUSION

k-MIP attention is a new scalable attention mechanism for graph transformers. It maintains the flexibility of full attention to propagate information between any two nodes, while achieving linear memory complexity and a ten-fold speedup over full attention in our implementation. Using k-MIP attention, we trained graph transformers on graphs with more than 500k nodes on a single 80GB A100 GPU. To our knowledge, this is the first time a non-linearized graph transformer with this flexibility has been demonstrated at such scale. Further, it consistently ranks among the top-performing scalable attention mechanisms on LRGB, City-Networks, ShapeNet-Part, and S3DIS. Finally, we have provided novel upper and lower bounds on the expressive power of graph transformers based on k-MIP attention. On one hand, we have proven that k-MIP transformers can approximate any full-attention transformer to arbitrary precision, guaranteeing that the sparsification does not come at the cost of model expressivity. On the other hand, we have proven that no graph transformer in the GraphGPS framework is more expressive than a specific instance of the SEG-WL test that depends on the used positional encoding.

**Limitations**    First, while k-MIP attention achieves significant practical speedups and scales to graphs with over 500k nodes, its computational complexity remains quadratic in the number of nodes, which may become prohibitive for extremely large graphs. Second, k-MIP attention inherently exposes each query to only a subset of keys at each step. This creates the possibility that if the most relevant keys for a particular query consistently fall outside the top-$k$ selection during training, the model may fail to learn the corresponding attention patterns (supervision starvation). One possible solution could be to start with a large $k$ and gradually decrease it as training progresses, allowing the model to learn from a wider range of keys in the beginning and then focus on the most relevant ones as it converges. Investigating the practical significance of this limitation and developing mitigation strategies remains an important direction for future work. Lastly, as discussed in Appendix K, pursuing low-level GPU optimizations similar to FlashAttention (Dao et al., 2022) would be paramount for its widespread adoption.

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

# A    BACKGROUND ON THE EXPRESSIVE POWER OF GRAPH NEURAL NETWORKS

In this appendix we provide additional literature review as well as mathematical background and definitions to complement the results in the main text.

## A.1    ADDITIONAL LITERATURE REVIEW ON THE EXPRESSIVE POWER OF GRAPH NEURAL NETWORKS

A lot of work has gone into overcoming the expressive power limitation of MPNNs, leading to the development of more expressive models like higher-order GNNs (Morris et al., 2019), as well as augmentations to the input features leading to higher expressive power. In early works, such augmentations took the form of unique and/or random node identifiers (Loukas, 2019; Sato et al., 2021), which unfortunately break the permutation invariance or equivariance of the GNN. Hence, more recent works have employed positional encodings based on eigenvectors of the adjacency matrix or Laplacian of the graph (Dwivedi et al., 2021; Lim et al., 2022), node and distance metrics (Li et al., 2020), substructure counts (Bouritsas et al., 2022), or random walks (Dwivedi et al., 2021). Many of these have been shown to lead to an expressive power beyond the 1-WL test. Incorporating positional encodings into the Weisfeiler-Lehman test leads to a preorder of WL test variations, where more discriminative encodings lead to a higher power for graph distinguishability. This preorder has been formalized in terms of the Structural Encoding Enhanced Global Weisfeiler-Lehman Test (SEG-WL test) from (Zhu et al., 2023).

In their work, (Zhu et al., 2023) also characterise the expressive power of various graph Transformer architectures in terms of a SEG-WL test, including the original graph Transformer (Dwivedi & Bresson, 2020), Graphormer (Ying et al., 2021), SAN (Kreuzer et al., 2021), Gophormer (Zhao et al., 2021), and SAT (Chen et al., 2022a). In this work, we extend their analysis by showing that the expressive power of the GraphGPS framework (including our own GPS+k-MIP, Exphormer (Shirzad et al., 2023), and GPS++ (Masters et al., 2022)) is upper bounded by a SEG-WL test with a structural encoding scheme that is determined by the input node features, the input edge features, and the positional encodings used in the model. By adopting the same framework, our result can be directly compared to those already established for the architectures listed above. In particular, we show that in the usual 1-WL setting, where all node and edge features are identical and there are no positional encodings, the GraphGPS framework cannot distinguish graphs that the 1-WL test cannot distinguish. We use this insight to advocate for the use of expressive positional encodings.

When sufficiently expressive positional encodings are used, however, many graph transformers can leverage the universal approximation property of sequence transformers to approximate any continuous function on graphs to arbitrary precision. Such a universal approximation result has been proven for SAN (Kreuzer et al., 2021) and Exphormer (Shirzad et al., 2023) under the assumption that a maximally expressive positional encoding (the padded adjacency matrix) is used. in this work, we will prove a completely analogous result for the GPS+k-MIP, showing that no expressivity is lost by using the k-MIP self-attention mechanism in graph transformers.

## A.2    NODE COLORINGS AND THE SEG-WL TEST

As discussed in section 4, node coloring algorithms such as the WL test have been used to establish important upper and lower bounds on the expressive power of GNNs. We contribute to this literature by showing a similar upper bound on the expressive power of the GraphGPS framework, using the SEG-WL test from Zhu et al. (2023). This section provides an overview of the background necessary to understand node coloring algorithms and the SEG-WL test.

### A.2.1    NODE COLORINGS AND THE 1-WL TEST

We denote a *multiset* of elements by $\{\{a_1, \ldots, a_n\}\}$, where the order of the elements does not matter. We denote the set of class of multisets with elements from a class $\mathcal{S}$ by $\mathbb{N}^{\mathcal{S}}$.

**Definition 3.** *A node coloring of a graph $G$ is a function $c : V \to \mathcal{C}$ that assigns a colour to each node in $G$. Here, $\mathcal{C}$ denotes the set of possible colours.*

Note that there is no restriction on the class $\mathcal{C}$. In particular, $\mathcal{C}$ could be $\mathbb{R}^d$, therefore any GNN and every instance of the GraphGPS framework can be seen as an algorithm that generates node colorings.

The Weisfeiler-Lehman test (abbreviated 1-WL test, to distinguish it from higher-order variations Morris et al. (2019)) is such a node coloring algorithm that was originally introduced to test graph isomorphism. The test iteratively refines the node coloring of a graph by aggregating the colours of neighbouring nodes, as described in the following definition and illustrated in fig. 5.

**Definition 4.** *The* Weisfeiler-Lehman Test *(1-WL Test) is a graph isomorphism test that iteratively refines the node coloring of a graph as described below.*

*Let the input be a graph $G = (V, E)$ with initial node coloring $c^0 : V \to \mathcal{C}$. The test iteratively updates the colour of each node $v \in V$ as*

$$c^l(v) = \tau\left(c^{l-1}(v), \{\{c^{l-1}(r) \mid r \in \mathcal{N}_{in}(v)\}\}\right), \tag{9}$$

*where $\tau$ is an injective map from $\mathcal{C} \times \mathbb{N}^{\mathcal{C}}$ to $\mathcal{C}$.*

*The algorithm can be terminated after a fixed number of iterations L, or when the equivalence classes of the node colorings stabilise.*

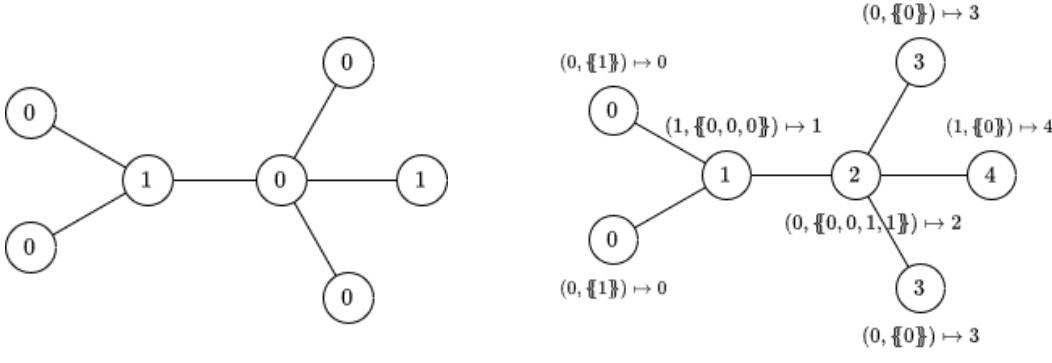

(a) Initial node coloring (iteration 0)                 (b) Node coloring after 1 iteration

Figure 5: Illustration of a single iteration of the Weisfeiler-Lehman test. The node labels (in $\mathcal{C} = \mathbb{N}$) are written inside the nodes. The mapping $\tau$ is written next to the nodes.

The 1-WL test can be used to test whether two graphs are isomorphic, by comparing the multisets of node colours generated by the test on the two graphs. If the multisets are different, the graphs are guaranteed to be non-isomorphic. In this case, the 1-WL test is said to distinguish both graphs. However, if the multisets are the same, they may or may not be isomorphic. This idea can be extended to other node coloring algorithms as follows.

**Definition 5.** *An algorithm $\mathcal{A}$ that generates node colorings* distinguishes *two non-isomorphic graphs $G_1$ and $G_2$ iff the multisets of node colorings generated by $\mathcal{A}$ on $G_1$ and $G_2$ are not equal, i.e. iff*

$$\{\{\mathcal{A}(G_1)(v) \mid v \in V_1\}\} \neq \{\{\mathcal{A}(G_2)(v) \mid v \in V_2\}\}. \tag{10}$$

While the 1-WL test is a powerful tool for distinguishing pairs of non-isomorphic graphs, it cannot distinguish all such pairs. Figure 6 shows an example of two non-isomorphic graphs that are indistinguishable by the 1-WL test (Rattan & Seppelt, 2023).

### A.2.2  SEG-WL TEST

Augmenting the input features with positional encodings can increase the power of graph neural networks for distinguishing non-isomorphic graphs. In Zhu et al. (2023), the authors introduce the Structural Encoding Enhanced Global Weisfeiler-Lehman Test (SEG-WL test), which is a generalisation of the 1-WL test that incorporates structural encodings into the node coloring algorithm. Because we will characterise the expressive power of the GraphGPS framework (which includes the

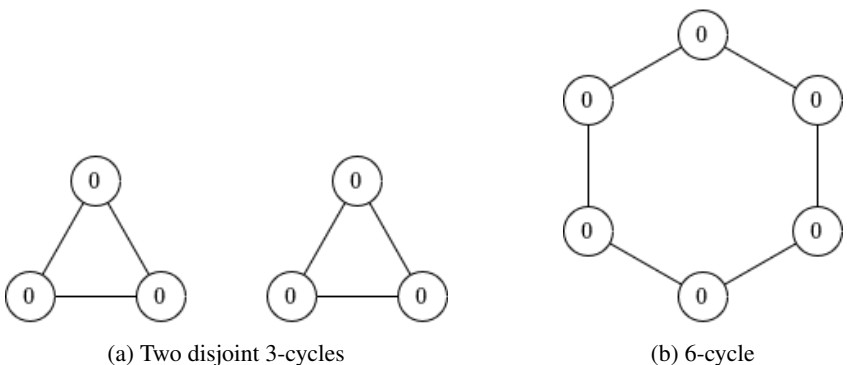

(a) Two disjoint 3-cycles                                         (b) 6-cycle

Figure 6: Two graphs that are indistinguishable by the 1-WL test.

GPS+k-MIP) using the SEG-WL test, we will define the SEG-WL test in the remainder of this section. The following definition forms a general framework for feature augmentation, that can augment both node-wise and edge-wise features.

**Definition 6** (Zhu et al. (2023)). *A structural encoding scheme $S = (f_A, f_R)$ is a pair of functions, where for any graph $G = (V, E)$, $f_A(v, G) \in \mathcal{C}$ defines the encoding of any node $v \in V$ and $f_R(v, u, G) \in \mathcal{C}$ defines the encoding of any node pair $(v, u) \in V \times V$. $S$ is called* strongly regular *if there exists a discriminator function $\xi : \mathcal{C} \times \mathcal{G} \to \{0, 1\}$ such that $\xi(f_R(v, u, G), G) = 1$ if and only if $u = v$.*

For each structural encoding scheme $S$, there is an associated $S$-SEG-WL test, which is defined as follows.

**Definition 7** (Zhu et al. (2023)). *The $S$-SEG-WL Test, where $S = (f_A, f_R)$ is a structural encoding scheme, is a graph isomorphism test that iteratively refines the node coloring of a graph as described below.*

*Let the input be a graph $G = (V, E, \boldsymbol{X}, \boldsymbol{E}, \boldsymbol{A})$. The test proceeds as follows:*

1. *Initialize the node coloring $c^0 : V \to \mathcal{C}$ as*

$$c^0(v) = \Phi_0(\boldsymbol{X}_v, f_A(v, G)), \tag{11}$$

   *where $\Phi_0$ is an injective function from $\mathbb{R}^d \times \mathcal{C}$ to $\mathcal{C}$.*

2. *In iteration $l$, compute the colour $c^l(v)$ of each node $v \in V$ as*

$$c^l(v) = \Phi(\{\{(c^{l-1}(r), f_R(v, r, G)) \mid r \in V\}\}), \tag{12}$$

   *where $\Phi$ is a function that injectively maps $\mathbb{N}^{\mathcal{C} \times \mathcal{C}}$ to $\mathcal{C}$.*

### A.2.3 THE SEG-WL PREORDER

Different node coloring algorithms can be compared in terms of their expressive power by comparing the pairs of graphs that they can distinguish. If an algorithm $\mathcal{A}$ can distinguish every pair of non-isomorphic graphs that $\mathcal{B}$ can distinguish, we say that $\mathcal{A}$ is *more expressive* than $\mathcal{B}$. If, in addition, there exists a pair of graphs that $\mathcal{A}$ can distinguish but $\mathcal{B}$ cannot, we say that $\mathcal{A}$ is *strictly more expressive* than $\mathcal{B}$. It is easy to check that the relation "is more expressive than" is reflexive and transitive, and therefore forms a preorder on the class of node coloring algorithms. Note that it is not antisymmetric, so it is not a partial order as was wrongly stated in Zhu et al. (2023).

As there is a SEG-WL test corresponding to each structural encoding scheme, the relation "is more expressive than" also forms a preorder on the class of SEG-WL tests. One result that provides insight in this preorder is Theorem 3 from Zhu et al. (2023), which embodies the intuitive result that SEG-WL tests become more expressive as their structural encoding schemes become more discriminative. To formalize this, we first introduce the notion of a *refinement* of a structural encoding scheme.

**Definition 8.** *Consider two structural encoding schemes $S = (f_A, f_R)$ and $S' = (f'_A, f'_R)$. We call $S'$ a* refinement *of $S$ (notated $S' \succsim S$) if there exist mappings $p_A, p_R$ such that for any $G$ and $u, v \in V$, we have*

$$f_A(v, G) = p_A(f'_A(v, G)), \tag{13}$$

$$f_R(v, u, G) = p_R(f'_R(v, u, G)). \tag{14}$$

*Then the $S$-SEG-WL test is more expressive than the $S'$-SEG-WL test in testing non-isomorphic graphs.*

**Theorem 3** (Theorem 3 in Zhu et al. (2023))**.** *For two structural encoding schemes $S$ and $S'$, if $S' \succsim S$, then the $S'$-SEG-WL test is more expressive than the $S$-SEG-WL test. Further, if $S$-SEG-WL distinguishes two non-isomorphic graphs $G_1$ and $G_2$ after $t$ iterations, then $S'$-SEG-WL distinguishes $G_1$ and $G_2$ after at most $t$ iterations.*

## B PROOF OF THEOREM 1

In this appendix, we provide the proof for Theorem 1, repeated here for clarity.

**Theorem 4** (Theorem 1 repeated)**.** *Let $\mathcal{A}$ be an instance of the GraphGPS framework that enhances its node features with $\nu(\boldsymbol{A})$ and its edge features with $\mu(\boldsymbol{A})$. Then $\mathcal{A}$ can only distinguish graphs that are also distinguishable by the $S$-SEG-WL test, where $S = (f_A, f_R)$ and $f_A, f_R$ are defined as*

$$f_A(v, G) = \nu(\boldsymbol{A})_v, \tag{15}$$

$$f_R(v, u, G) = \begin{cases} (0, \mathbb{1}_{u=v}, \boldsymbol{0}_{d_{edge}}, \boldsymbol{0}_{d_{PE}}) & \text{if } (u, v) \notin E \\ (1, \mathbb{1}_{u=v}, \boldsymbol{E}_{uv}, \mu(\boldsymbol{A})_{uv}) & \text{if } (u, v) \in E \end{cases}, \tag{16}$$

*where the set of possible colors is $\mathcal{C} = \mathbb{R}^{d_{PE}} \cup \{0, 1\}^2 \times \mathbb{R}^{d_{edge}} \times \mathbb{R}^{d_{PE}} \cup \mathbb{R}^d \cup \mathbb{R}^{d_{edge}}$.*

For this theorem, we slightly simplify the GraphGPS architecture presented in Figure 1 by omitting batch normalization. This simplification is necessary to ensure that the output of the model depends only on a single input graph, which enables the characterization of the model as a mapping. This mapping is also deterministic, because dropout is disabled at inference time. Further, we focus on node-level tasks, where no graph pooling or edge decoding is applied; instead, the node features are directly returned without additional processing. The results we present are, however, easily extensible to graph- and edge-level tasks.

We would like to highlight that we use the following properties of the components of the GraphGPS framework. Note that assuming these properties is not restrictive, as they are satisfied by all instances of MPNN, Attn, and MLP implemented in the GraphGPS framework. In particular, we do not yet assume that Attn is the k-MIP self-attention mechanism, as our results in Section 4.1 hold for any instance of the GraphGPS framework.

A1. The output embedding of MPNN for node $v$ only depends on the previous embedding of $v$, the embeddings of the nodes in the in-neighbourhood of $v$ and the edge embeddings of the corresponding incoming edges. Further, it is invariant w.r.t. permutations of the neighbours.

A2. The output embedding of MPNN for edge $(r, v)$ only depends on the previous embedding of $(r, v)$ and the previous embeddings of the nodes $r$ and $v$.

A3. Attn is equivariant w.r.t. token permutations, and thus to permutations of the node indices.

A4. MLP acts on each node embedding independently.

### B.1 PRELIMINARY LEMMA

To prove Theorem 1, we will first establish the following lemma. This lemma and its proof are based on Theorem 1 in Zhu et al. (2023), albeit substantial modifications were necessary to account for the fact that the GraphGPS framework iteratively computes edge embeddings in addition to node embeddings.

**Lemma 1** (Theorem 1 in Zhu et al. (2023), modified). *For any strongly regular structural encoding scheme $S = (f_A, f_R)$ and graph $G = (V, E)$ with node features $\boldsymbol{X}$, if a graph neural model $\mathcal{A}$ that computes node embeddings $d^l : V \rightarrow \mathcal{C}$ and node pair embeddings $e^l : V \times V \rightarrow \mathcal{C}$ satisfies the following conditions:*

C1. *$\mathcal{A}$ computes the initial node and node pair embeddings with*

$$d^0(v) = \phi(\boldsymbol{X}_v, f_A(v, G)) \tag{17}$$

$$e^0(r, v) = \psi(f_R(v, r, G)) \tag{18}$$

*where $\phi$ and $\psi$ are model-specific functions.*

C2. *$\mathcal{A}$ aggregates and updates node and edge embeddings iteratively with*

$$d^l(v) = \sigma\left(\{\{(d^{l-1}(r), e^{l-1}(r, v), f_R(v, r, G)) \mid r \in V\}\}\right) \tag{19}$$

$$e^l(r, v) = \tau\left(e^{l-1}(r, v), d^{l-1}(r), d^{l-1}(v), f_R(v, r, G)\right) \tag{20}$$

*where $\sigma$ and $\tau$ are model-specific functions.*

*Then any two graphs $G_1$ and $G_2$ that are distinguished by $\mathcal{A}$ in iteration $l$ are also distinguished by S-SEG-WL in iteration $l$.*

*Proof.* We consider two not necessarily distinct graphs $G_v = (V_v, E_v, \boldsymbol{X}_v, \boldsymbol{E}_v, \boldsymbol{A}_v)$ and $G_w = (V_w, E_w, \boldsymbol{X}_w, \boldsymbol{E}_w, \boldsymbol{A}_w)$ and denote the colour given by S-SEG-WL to node $u \in V_v \cup V_w$ at iteration $l$ by $c^l(u)$. We first show by induction that for any iteration $l$ and any nodes $v, r \in V_v, w, s \in V_w$, the following two implications hold:

$$c^l(v) = c^l(w) \implies d^l(v) = d^l(w) \tag{IH.1}$$

$$\left. \begin{array}{r} c^l(r) = c^l(s) \\ f_R(v, r, G_v) = f_R(w, s, G_w) \\ c^l(v) = c^l(w) \end{array} \right\} \implies e^l(r, v) = e^l(s, w) \tag{IH.2}$$

**Base case** For $l = 0$, eq. (IH.1) holds because $c^0(v) = c^0(w)$ implies $\boldsymbol{X}_v = \boldsymbol{X}_w$ and $f_A(v, G_v) = f_A(w, G_w)$, which leads to $d^0(v) = d^0(w)$. Further, eq. (IH.2) holds because $f_R(v, r, G_v) = f_R(w, s, G_w)$ directly entails $e^0(r, v) = e^0(s, w)$.

**Induction step for eq. (IH.1)** Suppose that the induction hypotheses eq. (IH.1) and eq. (IH.2) hold for iteration $l$ and that $c^{l+1}(v) = c^{l+1}(w)$. From the injectivity of function $\Phi$, we have

$$\{\{(c^l(r), f_R(v, r, G_v)) \mid r \in V_v\}\} = \{\{(c^l(s), f_R(w, s, G_w)) \mid s \in V_w\}\} \tag{21}$$

Because $f_R$ is strongly regular, we can choose a discriminator function $\xi$ such that for all graphs $G$ and nodes $a, b, \xi(f_R(a, b, G), G) = \mathbb{1}_{a=b}$. By applying $\xi$ to the second element of each tuple in both multisets together with the corresponding graph, we obtain

$$\{\{(c^l(r), \mathbb{1}_{r=v}) \mid r \in V_v\}\} = \{\{(c^l(s), \mathbb{1}_{s=w}) \mid s \in V_w\}\} \tag{22}$$

In each of the above multisets, there is a unique tuple for which the second element is 1, namely the tuples corresponding to $r = v$ and $s = w$, respectively. Therefore, these tuples must be equal, which implies

$$c^l(v) = c^l(w) \tag{23}$$

Because the two multisets in eq. (21) are identical and finite, their elements can be matched in pairs. Further, by the induction hypotheses and the fact that $c^l(v) = c^l(w)$, we have that for any $r \in V_v$ and $s \in V_w$, $(c^l(r), f_R(v, r, G_v)) = (c^l(s), f_R(w, s, G_w))$ implies $(d^l(r), e^l(r, v), f_R(v, r, G_v)) = (d^l(s), e^l(s, w), f_R(w, s, G_w))$. Hence,

$$\begin{aligned} \{\{(d^l(r), e^l(r, v), f_R(v, r, G_v))\}\} \mid r \in V_v) \\ = \{\{(d^l(s), e^l(s, w), f_R(w, s, G_w)) \mid s \in V_w\}\} \end{aligned} \tag{24}$$

Considering $\mathcal{A}$ updates node labels according to eq. (19), $d^{l+1}(v) = d^{l+1}(w)$ holds. This completes the induction step for eq. (IH.1).

**Induction step for eq. (IH.2)**    While retaining our assumptions that the induction hypotheses eq. (IH.1) and eq. (IH.2) hold for iteration $l$ and that $c^{l+1}(v) = c^{l+1}(w)$, suppose additionally that $c^{l+1}(r) = c^{l+1}(s)$ and $f_R(v, r, G_v) = f_R(w, s, G_w)$. Analogously to the derivation preceding eq. (23), it follows from $c^{l+1}(v) = c^{l+1}(w)$ that $c^l(v) = c^l(w)$ and from $c^{l+1}(r) = c^{l+1}(s)$ that $c^l(r) = c^l(s)$. Induction hypothesis eq. (IH.1) now yields $d^l(v) = d^l(w)$ and $d^l(r) = d^l(s)$, while induction hypothesis eq. (IH.2) leads to $e^l(r, v) = e^l(s, w)$. Combining these results, we have

$$
\begin{aligned}
&(e^l(r, v), d^l(r), d^l(v), f_R(v, r, G_v)) \\
=&(e^l(s, w), d^l(s), d^l(w), f_R(w, s, G_w))
\end{aligned}
\tag{25}
$$

And because $\mathcal{A}$ updates edge labels according to eq. (20), we have $e^{l+1}(r, v) = e^{l+1}(s, w)$. This completes the induction step for eq. (IH.2), and thereby the proof that eqs. (IH.1) and (IH.2) hold for all iterations $l$.

**Consequence of eq. (IH.1)**    Now consider two graphs $G_1$ and $G_2$ that are distinguished by $\mathcal{A}$ after $l$ iterations. We will prove by contradiction that $G_1$ and $G_2$ are also distinguished by the $S$-SEG-WL test after $l$ iterations. Suppose that $G_1$ and $G_2$ are not distinguished by the $S$-SEG-WL test after $l$ iterations, i.e.

$$
\{\!\{c^l(v) \mid v \in V_1\}\!\} = \{\!\{c^l(w) \mid w \in V_2\}\!\}
\tag{26}
$$

Because the two multisets in eq. (26) are identical and finite, their elements can be matched in pairs of equal elements. By eq. (IH.1), we have that for any $v \in V_1, w \in V_2$, $c^l(v) = c^l(w)$ implies $d^l(v) = d^l(w)$, from which

$$
\{\!\{d^l(v) \mid v \in V_1\}\!\} = \{\!\{d^l(w) \mid w \in V_2\}\!\}
\tag{27}
$$

This would imply that $G_1$ and $G_2$ are not distinguished by $\mathcal{A}$ after $l$ iterations, which contradicts our assumption. Therefore, $G_1$ and $G_2$ must be distinguished by the $S$-SEG-WL test after $l$ iterations.  $\square$

Now that Lemma 1 is proven, we can proceed with the proof of Theorem 1.

## B.2    Reduction of Theorem 1 to Lemma 1

*Proof.* Let $\mathcal{B}$ be an instance of the GraphGPS framework (section 4.1) that iteratively generates the node and edge embeddings $(\boldsymbol{X}^l)_{l=0}^L$ and $(\boldsymbol{E}^l)_{l=0}^L$. Extend $\mathcal{B}$ to the coloring algorithm $\mathcal{A}$ that iteratively computes node and node pair colours as follows:

$$
d^l(v) = \boldsymbol{X}_v^l
\tag{28}
$$

$$
e^l(r, v) = \begin{cases} \boldsymbol{0}_d & \text{if } (r, v) \notin E \\ \boldsymbol{E}_{rv}^l & \text{if } (r, v) \in E \end{cases}
\tag{29}
$$

First, note that the structural encoding scheme in the theorem statement is strongly regular, as any function $\xi$ for which $\xi(t, G) := t.\texttt{second}$ if $t$ is a 4-tuple is a discriminator function for $f_R$. We will now prove that $\mathcal{A}$ satisfies the conditions of Lemma 1, where $S = (f_A, f_R)$ and $\mathcal{C}$ are as defined in the theorem statement. Once this is established, it follows that $\mathcal{A}$ can only distinguish graphs that are also distinguishable by the $S$-SEG-WL test. The desired result then follows from the observation that $\mathcal{A}$ and $\mathcal{B}$ always generate the same node colorings, and thus distinguish exactly the same graphs.

$\mathcal{A}$ satisfies Condition C1 by construction: the initial node and node pair colours are computed as

$$
d^0(v) = \boldsymbol{X}_v^0 = \begin{bmatrix} \boldsymbol{X}_v \\ \nu(\boldsymbol{A})_v \end{bmatrix} = \phi(\boldsymbol{X}_v, f_A(v, G))
\tag{30}
$$

$$
e^0(r, v) = \begin{cases} \boldsymbol{0}_d & \text{if } (r, v) \notin E \\ \begin{bmatrix} \boldsymbol{E}_{rv} \\ \mu(\boldsymbol{A})_{rv} \end{bmatrix} & \text{if } (r, v) \in E \end{cases} = \psi(f_R(v, r, G))
\tag{31}
$$

if $\phi$ is chosen to be the concatenation operation and $\psi$ is chosen to be a function that concatenates the last two elements when given a 4-tuple.

$\mathcal{A}$ also satisfies Condition C2. To see this, we will describe the construction of the functions $\sigma$ and $\tau$ such that the update rule of $\mathcal{A}$ can be written as eqs. (19) and (20).

$\sigma$ takes in $I_v = \{\{(\boldsymbol{X}_r^{l-1}, e^{l-1}(r,v), f_R(v,r,G)) \mid r \in V\}\}$ and outputs $d^l(v) = \boldsymbol{X}_v^l$. Given the input $I_v$, first retrieve the node feature vector $\boldsymbol{X}_v^{l-1}$ by selecting the unique tuple $t \in I_v$ for which $\mathbb{1}_{r=v} = f_R(v,r,G).\texttt{second} = 1$ and letting $\boldsymbol{X}_v^{l-1} = t.\texttt{first}$. Then, retrieve the multiset $\{\{(\boldsymbol{X}_r^{l-1}, \boldsymbol{E}_{rv}^{l-1}) \mid (r,v) \in E\}\}$ from $I_v$ by selecting $\boldsymbol{X}_r^{l-1}$ and $\boldsymbol{E}_{rv}^{l-1}$ from all tuples $t$ for which $\mathbb{1}_{(r,v)\in E} = f_R(v,r,G).\texttt{first} = 1$. Because of Assumption A1, we now have all the necessary inputs to apply the message passing layer MPNN and obtain $\boldsymbol{X}_{M,v}^p lus{*}l$. Further, compute $\boldsymbol{X}_{M,v}^l = \boldsymbol{X}_{M,v}^p lus{*}l + \boldsymbol{X}_v^{l-1}$. Next, retrieve the multiset $\{\{\boldsymbol{X}_r^{l-1} \mid r \in V\}\}$ from $I_v$ by selecting the first element of all tuples. Assumption A3 guarantees that this and $\boldsymbol{X}_v^{l-1}$ is all we need to determine $\mathrm{Attn}(\boldsymbol{X}^{l-1})_v$. Thus, we can compute $\boldsymbol{X}_{T,v}^l = \mathrm{Attn}(\boldsymbol{X}^{l-1})_v + \boldsymbol{X}_v^{l-1}$. Finally, Assumption A4 ensures that MLP computes an elementwise function, so we can compute $\boldsymbol{X}_v^l = \mathrm{MLP}(\boldsymbol{X}_{M,v}^l + \boldsymbol{X}_{T,v}^l) + \boldsymbol{X}_{M,v}^l + \boldsymbol{X}_{T,v}^l$. By following this procedure, $\sigma$ can implement the desired update rule.

$\tau$ takes in $I_{rv} = (e^{l-1}(r,v), d^{l-1}(r), d^{l-1}(v), f_R(r,v,G))$ and outputs $e^l(r,v)$. Given the input $I_{rv}$, first determine whether $(r,v) \in E$ by checking the third element of $f_R(v,r,G)$. If $(r,v) \notin E$, output $\boldsymbol{0}_d$. Otherwise, Assumption A2 guarantees that $e^l(r,v) = \boldsymbol{E}_{rv}^l$ depends only on $e^{l-1}(r,v)$, $d^{l-1}(r)$ and $d^{l-1}(v)$. Thus, $\tau$ can be implemented as desired.

This completes the proof that $\mathcal{A}$ satisfies the conditions of Lemma 1, where $S = (f_A, f_R)$ and $\mathcal{C}$ are as defined in the theorem statement. It follows that $\mathcal{A}$ can only distinguish graphs that are also distinguishable by the $(f_A, f_R)$-SEG-WL test. Now observe that $\mathcal{A}$ and $\mathcal{B}$ always generate the same node colorings, and thus distinguish exactly the same graphs. From this follows the desired result. $\qquad\square$

## C  PROOF OF THEOREM 2

To prove Theorem 2, we first establish the following lemma.

**Lemma 2.** *Consider any compact set $U \subseteq \mathbb{R}^{N \times d}$ and any function $f : U \to \mathbb{R}^{N \times d}$ that satisfies the following conditions:*

- *$f$ is continuous w.r.t. any entry-wise $\ell^p$ norm with $p \in [1, \infty)$.*

- *$f$ is equivariant w.r.t. row permutations, i.e. for any permutation matrix $\boldsymbol{P}$ such that $f(\boldsymbol{X})$ and $f(\boldsymbol{P}\boldsymbol{X})$ are well-defined we have*

$$f(\boldsymbol{P}\boldsymbol{X}) = \boldsymbol{P}f(\boldsymbol{X}). \tag{32}$$

*Then there exists a k-MIP transformer $T_{k\text{-}MIP} \in \mathcal{T}_{k\text{-}MIP}^{2,1,4}$ such that*

$$\left( \int_U ||f(\boldsymbol{X}) - T_{k\text{-}MIP}(\boldsymbol{X})||_p^p d\boldsymbol{X} \right)^{1/p} < \epsilon. \tag{33}$$

*Proof.* Yun et al. (Yun et al., 2019) provided a constructive proof of this theorem for full-attention transformers $T_{full} \in \mathcal{T}_{full}^{2,1,4}$ (Theorem 2 in their paper). Our proof is completely analogous; the only difference that needs to be accounted for is the substitution of $\mathrm{softmax}$ with $\mathrm{softmax} \circ \mathcal{T}_k$ in the attention mechanism.

One can note that the only property of the row-wise softmax function $\mathrm{softmax}$ that is used in the proof by Yun et al. (Yun et al., 2019) is that the output can be made arbitrarily close to a row-wise hardmax function $\mathrm{hardmax}$ by scaling up the input matrix $\boldsymbol{Z}$ by a factor $\lambda$:

$$\mathrm{softmax}(\lambda\boldsymbol{Z}) \to \mathrm{hardmax}(\lambda\boldsymbol{Z}) \quad \text{as} \quad \lambda \to \infty. \tag{34}$$

This property is also satisfied by $\mathrm{softmax} \circ \mathcal{T}_k$, which validates the proof for k-MIP transformers. $\quad\square$

Now we can proceed with the proof of Theorem 2.

*Proof of Theorem 2.* Consider any full-attention transformer $T_{full} \in \mathcal{T}_{full}^{h,m,r}$ and any compact set $U \subseteq \mathbb{R}^{N \times d}$. Let $f : U \to \mathbb{R}^{N \times d}$ be the restriction of $T_{full}$ to the domain $U$. Then we claim that $U$ and $f$ satisfy the conditions of Lemma 2:

- $T_{full}$ is a sequential composition of components that are continuous w.r.t. every norm in $\{\ell^p \mid p \in [1, \infty)\}$ on their entire domain. Therefore, $T_{full}$ is itself continuous w.r.t. every such norm. Consequently, its restriction $f$ is continuous w.r.t. every norm in $\{\ell^p \mid p \in [1, \infty)\}$.

- $T_{full}$ is a sequential composition of components that are equivariant w.r.t. row permutations. Hence, $T_{full}$ is itself equivariant w.r.t. row permutations. Consequently, its restriction $f$ is equivariant w.r.t. row permutations, i.e. for any permutation matrix $\boldsymbol{P}$ such that $f(\boldsymbol{X})$ and $f(\boldsymbol{PX})$ are well-defined we have

$$f(\boldsymbol{PX}) = \boldsymbol{P}f(\boldsymbol{X}). \tag{35}$$

Lemma 2 then guarantees that there exists a k-MIP transformer $T_{\text{k-MIP}} \in \mathcal{T}_{\text{k-MIP}}^{2,1,4}$ such that

$$\left( \int_U \|f(\boldsymbol{X}) - T_{\text{k-MIP}}(\boldsymbol{X})\|_p^p d\boldsymbol{X} \right)^{1/p} < \epsilon. \tag{36}$$

Since $T_{full}$ and $f$ are equal on $U$, it follows that for this k-MIP transformer $T_{\text{k-MIP}}$,

$$\left( \int_U \|T_{full}(\boldsymbol{X}) - T_{\text{k-MIP}}(\boldsymbol{X})\|_p^p d\boldsymbol{X} \right)^{1/p} < \epsilon. \tag{37}$$

$\square$

# D    DISCUSSION OF THEOREM 2

Theorem 2 establishes that any full-attention transformer can be approximated to arbitrary accuracy by a k-MIP transformer on any compact set of inputs. This fundamental result guarantees that the sparsification introduced by k-MIP attention does not compromise the expressive power of the model class. In this section, we discuss both what this theorem guarantees and its scope.

## D.1    KEY GUARANTEES

**Preservation of expressive power**    The sparsification inherent in k-MIP attention mechanisms does not fundamentally limit the class of functions that can be represented. Any function representable by a full-attention transformer can be approximated by a k-MIP transformer.

**Bounded architectural requirements**    Remarkably, the approximating k-MIP transformer has fixed architectural constraints: it belongs to $\mathcal{T}_{\text{k-MIP}}^{2,1,4}$, meaning it requires only 2 heads of dimension 1 and MLPs of dimension 4, regardless of the complexity of the target full-attention transformer (the universal construction is not necessarily practical). However, the theorem does not constrain the number of layers required, which may need to be arbitrarily large to achieve the desired approximation accuracy for complex target functions.

## D.2    SCOPE AND LIMITATIONS

The theorem has a specific scope that practitioners should understand to avoid potential misunderstandings.

**No guarantees on internal representations**    The theorem considers the approximation of the *function* (i.e. input-output behavior) $T_{full} : U \to \mathbb{R}^{N \times d}$ implemented by a full-attention transformer. This does not guarantee that the approximating k-MIP transformer will have intermediate representations or attention matrices that are similar to those of the full-attention transformer.

**No uniqueness of approximation** The theorem guarantees that there exists at least one k-MIP transformer that approximates the target function, where the proof provides a construction based on tiling the input space. However, this approximating k-MIP transformer is not necessarily the only one: an alternative approximation may be found in practice that is not captured by this construction.

**No optimization guarantees** The theorem guarantees that there *exists* at least one k-MIP transformer that approximates the target function. However, it does not guarantee that standard optimization algorithms (e.g., gradient descent) will find the exact approximation constructed by the proof, nor does it guarantee that any approximation is guaranteed to be found. The optimization landscape may present challenges that prevent convergence to an approximation of the full-attention transformer, or the full-attention transformer may be suboptimal for the task for which the k-MIP transformer is being optimized.

**No guarantee of equivalent expressive power for fixed-depth architectures** A corollary of the theorem is that the *class* of k-MIP transformers is equally expressive as the class of full-attention transformers. However, this does not guarantee that a k-MIP transformer with a specific number of layers will have equivalent expressive power to a full-attention transformer with the same depth.

**Compact domain restriction** The approximation guarantee only holds on compact sets. For unbounded domains, the theorem provides no guarantees about approximation quality. Nevertheless, this limitation has minimal practical impact, as input features in most real-world applications are typically normalized or naturally bounded in their range of values.

## D.3 PRACTICAL IMPLICATIONS

These theoretical guarantees suggest that k-MIP attention mechanisms are fundamentally sound from an expressivity standpoint, but practitioners should not expect automatic approximation or automatic performance parity with full attention. The gap between theoretical possibility and practical realizability depends on factors not addressed by the theorem, including optimization dynamics and finite-sample effects.

## D.4 RELATION TO PRIOR WORK

Theorem 2 builds heavily on Theorem 2 from (Yun et al., 2019), which states that any full-attention transformer (without positional encodings) is a universal function approximator of permutation-equivariant functions from $\mathbb{R}^{N \times d}$ to $\mathbb{R}^{N \times d}$. In our work, we extend this result to the case of k-MIP transformers. Lemma 2 is the equivalent of Theorem 2 from (Yun et al., 2019), where we show that any k-MIP transformer is a universal function approximator of permutation-equivariant functions from $\mathbb{R}^{N \times d}$ to $\mathbb{R}^{N \times d}$. Theorem 2 subsequently narrows the scope of the theorem to the approximation of full-attention transformers, as this is more relevant for the practical applications of k-MIP transformers.

Another related work is (Yun et al., 2020), which proposes a unified framework for sparse transformers and delineates conditions under which such sparse transformers are universal approximators of sequence-to-sequence functions. However, their approach differs from ours in three critical ways:

- Yun et al. (2020) deal with the approximation of general sequence-to-sequence functions, whereas our focus on the graph domain requires the approximation of functions that are equivariant w.r.t. node permutations. In Lemma 2, we proved that k-MIP transformers can indeed approximate all such permutation equivariant functions.

- The main result (Theorem 1) of Yun et al. (2020) deliberately breaks the natural permutation equivariance of transformers by requiring input features to be enhanced to $\boldsymbol{X'} = \boldsymbol{X} + \boldsymbol{E}$, where $\boldsymbol{E} \in \mathbb{R}^{N \times d}$ is a trainable positional embedding that breaks symmetry. In contrast, our work leverages this permutation equivariance as a fundamental desideratum for graph-based learning tasks, eliminating the need of such trainable embeddings and more closely resembling practical settings.

- We prove that all functions representable by a full-attention transformer satisfy the conditions of Lemma 2, proving that each full-attention transformer can be universally approximated.

We note that—apart from these three differences—k-MIP transformers would into the framework of (Yun et al., 2020) as a special case of sparse transformers with:

- $p = 1$: no cycling between sparsity patterns; there is only one sparsity pattern.

- $A_1^{(t)} = [n]$: the single sparsity pattern does not restrict which keys can attend to each other.

- Corollary to the above: $S_1^{(t)} = [n]$ for every $t \in \mathbb{N}$.

- $\rho(A) := \mathbf{softmax}(\tau_A(A))$.

However, as stated before, we believe this result to be less relevant for graph-based learning tasks where permutation equivariance is a fundamental desideratum.

## E  THE K-MIP ALGORITHM

---
**Algorithm 1** A Single Layer of Multi-Head k-MIP Self-Attention using Symbolic Matrices
---
**Require:** Input features $\boldsymbol{X} \in \mathbb{R}^{N \times d}$;
 1: Learnable matrices $\boldsymbol{W}_Q^h, \boldsymbol{W}_K^h \in \mathbb{R}^{d \times d_K}, \boldsymbol{W}_V^h \in \mathbb{R}^{d \times d_V}, \boldsymbol{W}_O^h \in \mathbb{R}^{d_V \times d}$
 2: number of keys to consider $k \in \mathbb{N}$, number of attention heads $H \in \mathbb{N}$
 3: **for** $h = 1$ to $H$ **do**
 4:     $\boldsymbol{S} \leftarrow \texttt{SymbolicMultiply}(\boldsymbol{X}\boldsymbol{W}_Q^h, (\boldsymbol{X}\boldsymbol{W}_K^h)^\top)$     $\triangleright \mathbb{R}^{N \times N}$, stored as symbolic matrix
 5:     $\boldsymbol{I}_{\text{argtopk}}, \boldsymbol{E}_{\text{topk}} \leftarrow \texttt{RowwiseTopK}(\boldsymbol{S}, k)$     $\triangleright \{0, 1, \ldots, N-1\}^{N \times k}, \mathbb{R}^{N \times k}$
 6:     $\boldsymbol{V}_{\text{topk}} \leftarrow \texttt{GatherRows}(\boldsymbol{X}\boldsymbol{W}_V^h, \boldsymbol{I}_{\text{argtopk}})$     $\triangleright \mathbb{R}^{N \times k \times d_V}$
 7:     $\boldsymbol{A}^h \leftarrow \text{softmax}(\boldsymbol{E}_{\text{topk}}/\sqrt{d_K})$     $\triangleright \mathbb{R}^{N \times k}$
 8:     $\boldsymbol{O}^h \leftarrow \texttt{Einsum}_{nk,nkv \to nv}(\boldsymbol{A}^h, \boldsymbol{V}_{\text{topk}})$     $\triangleright \mathbb{R}^{N \times d_V}$
 9: **end for**
10: $\boldsymbol{O} \leftarrow \sum_{h=1}^H \boldsymbol{O}^h \boldsymbol{W}_O^h$     $\triangleright \mathbb{R}^{N \times d}$
---

The full algorithm in pseudocode is presented in algorithm 1. For each head, the computation proceeds in three steps. First, lines 4-5 compute the indices of the $k$ highest entries in each row of $\boldsymbol{QK}^\top$ using symbolic matrices (Feydy et al., 2020), resulting in a matrix $\boldsymbol{I}_{\text{argtopk}} \in \mathbb{N}^{N \times k}$ of the top $k$ key indices and a matrix $\boldsymbol{E}_{\text{topk}} \in \mathbb{R}^{N \times k}$ of the corresponding inner products. Second, line 6 extracts the value vectors at these indices[3] and line 7 computes the attention scores by performing a row-wise softmax on $\boldsymbol{E}_{\text{topk}}$. Finally, line 8 computes the output for this head by summing the value vectors weighted by the attention scores. The final output is computed by summing the outputs of all heads.

Line 4 is the operation that requires the computation of $N^2$ inner products, and thus becomes the bottleneck as $N$ grows due to its quadratic complexity. However, thanks to the use of symbolic matrices using Kernel Operations (KeOps) (Charlier et al., 2021), these intermediate inner products are never materialized in the GPU's global memory and instead computed lazily in CUDA registers, thus avoiding memory overflows and high-latency memory transfers. This innovation is key to the scalability of k-MIP attention: despite retaining a quadratic computational complexity, it gives the algorithm a linear memory footprint and a speedup of an order of magnitude, which we empirically confirm in section 5.1. Furthermore, the matrices $\boldsymbol{E}_{\text{argtopk}}$ and $\boldsymbol{E}_{\text{topk}}$ do not require recomputation during backpropagation, rendering the backward pass computationally negligible relative to the forward pass when $N$ is large.

While maximum inner product search is a well-studied problem in the field of information retrieval and recommendation systems (Shrivastava & Li, 2014), the techniques used in those fields were not beneficial in our method. In particular, we conducted preliminary experiments with IVF, PQ, and LSH indexes in FAISS (Johnson et al., 2019), but these approaches failed to deliver a speedup that justified the associated loss in recall.

---

[3]The $\texttt{GatherRows}$ operation results in $\boldsymbol{V}_{\text{topk}}[n, k', :] = (\boldsymbol{X}\boldsymbol{W}_V^h)[\boldsymbol{I}_{\text{argtopk}[n, k']}, :]$ for all indices $n, k'$

# F  SYMBOLIC MATRICES

Traditionally in machine learning, matrices are stored as either dense or sparse matrices. Both of these methods store each element in memory at a known location as depicted in figs. 7a and 7b, respectively. When there are many non-zero elements, however, both have a large memory footprint and slow storage and retrieval times.

Symbolic matrices, as popularised by Feydy et al. (Feydy et al., 2020), take a different approach. Instead, they store the elements of the matrix as a formula $\boldsymbol{M}_{i,j} = F(\boldsymbol{x}_i, \boldsymbol{y}_j)$ that is evaluated on data arrays $(\boldsymbol{x}_i)_{i=1}^N$ and $(\boldsymbol{y}_j)_{j=1}^M$. Reduction operations are evaluated lazily, with high levels of parallelism, and computed without ever sending intermediate results to the GPU's global memory, which can make them 30 to 1000 times more efficient than their dense counterparts (Feydy et al., 2020).

In this work, we use symbolic matrices to compute, for all queries $\boldsymbol{q}_i$, the $k$ keys $\boldsymbol{k}_j$ that have the highest inner products with $\boldsymbol{q}_i$. We first define the symbolic matrix $\boldsymbol{E}$ with the formula $\boldsymbol{E}_{i,j} = \boldsymbol{q}_i \cdot \boldsymbol{k}_j$. Then, we apply a reduction on the rows of $\boldsymbol{E}$ that computes the indices of the $k$ largest elements of each row.

Because of our use of symbolic matrices, this computation never materialises the results of the $N^2$ inner products in GPU main memory and instead makes use of the GPU's shared memory and registers. The result is that our implementation of k-MIP self-attention has a negligible memory footprint for both the forward and backward pass, and that it is an order of magnitude faster than its implementation with dense matrices. For a comparison with FlashAttention (Dao et al., 2022) refer to Appendix K.

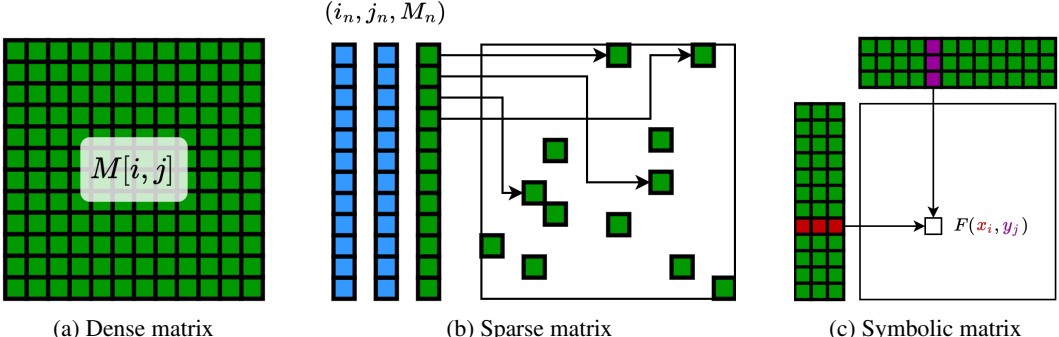

|                    |                    |                       |
| :----------------: | :----------------: | :-------------------: |
| (a) Dense matrix   | (b) Sparse matrix  | (c) Symbolic matrix   |

Figure 7: A comparison of dense, sparse, and symbolic matrices. Based on Feydy et al. (Feydy et al., 2020).

# G  ADDITIONAL RELATED WORK

## G.1  COMPARISON BETWEEN K-MIP ATTENTION AND OTHER SCALABLE GRAPH TRANSFORMERS

In the categorization of section 2.2, we have positioned k-MIP attention within the *learnable patterns* category. This is because k-MIP attention, like other methods in this category, allows the model to dynamically determine which nodes can attend to each other. It does this without imposing any restrictions beyond $k$-regularity on potential node interactions. This flexibility is an advantage over most existing approaches that employ graph subsampling or engineered attention patterns, as it may enable k-MIP attention to capture dependencies that are missed by more restrictive attention mechanisms.

Regarding scalability, k-MIP attention shares the linear memory complexity of most existing scalable graph transformers, making it suitable for processing large graphs. While its time complexity remains quadratic, our implementation with symbolic matrices provides significant practical efficiency. This allows k-MIP attention to scale to graphs with approximately 500,000 nodes at a computational

budget roughly 2 times slower than Performer and 3 times faster than BigBird, as demonstrated in our experiments on the City-Networks benchmark in section 5.3 and Appendix J.2.

This scale of operation (hundreds of thousands of nodes) is consistent with the current capabilities of most state-of-the-art scalable graph transformers. Models leveraging linearized approximations of attention, such as NodeFormer (Wu et al., 2022), and Difformer (Wu et al., 2023), have been demonstrated on graphs with up to a few million nodes. GOAT (Kong et al., 2023), which performs full attention w.r.t. k-means clusters of the keys, attains a similar scale. The notable exception is SGFormer (Wu et al., 2024), which has been trained on much larger graphs such as arxiv-100M. Note however that the "attention" mechanism in the latter is closer to a linear projection layer than to a true attention mechanism.

## G.2    COMPARISON WITH SPEXPHORMER

Additionally, while we do not consider SpExphormer (Shirzad et al., 2024) to be the most directly related work to our approach, we provide a tentative comparison to clarify the differences in modeling assumptions, training procedures, and theoretical guarantees.

At a high level, k-MIP attention replaces full self-attention with a k-Maximum Inner Product (k-MIP) operator: for each query node, only the top-$k$ keys according to inner product are retained. This sparsification is performed implicitly using symbolic matrix primitives (e.g., KeOps), so full attention matrices are never materialized in GPU memory. In contrast, SpExphormer follows a two-stage pipeline. First, a narrow Exphormer-style model is trained to learn attention scores over a fixed computational graph. Second, for each layer, only a small fixed number of highest-scoring edges is retained, and a wider model is retrained on the resulting sparse graph.

The induced computational graphs also differ substantially. In k-MIP attention, each layer and each attention head induces its own $k$-regular directed graph, with no restrictions on which nodes may attend to one another. SpExphormer, by contrast, uses a fixed computational graph during its first training stage, consisting of the input graph augmented with an expander graph (without virtual nodes). During the second stage, a new sparse $k$-regular directed graph is sampled at every epoch and layer via reservoir sampling, based on the learned attention scores over the Stage 1 graph.

From a computational perspective, k-MIP attention retains the worst-case quadratic time complexity of full attention, though with reduced constants due to top-$k$ selection and without materializing dense attention matrices. Its memory complexity scales linearly with the number of nodes and $k$. SpExphormer's complexity depends on the training stage: Stage 1 has costs comparable to Exphormer, while Stage 2 scales linearly with the number of retained edges, resulting in significantly reduced memory usage during wide-model training.

The two approaches also differ in their theoretical guarantees. For k-MIP attention, we establish an approximation theorem showing that, for any full-attention transformer and any compact set, there exists a shallow k-MIP transformer that can approximate it arbitrarily well. SpExphormer provides complementary guarantees: one result shows that sufficiently narrow networks can approximate arbitrarily wide transformers under boundedness assumptions, while another bounds the spectral norm error introduced by edge sampling with high probability.

Empirically, the largest graph processed with k-MIP attention is the London road network, containing 569k nodes and 759k edges, trained on a single 80GB A100 GPU. SpExphormer demonstrates scalability to larger graphs, such as Amazon2M with two million nodes, relying on substantial CPU memory (500GB RAM) and a 40GB A100 GPU while using only a small fraction of GPU memory.

Overall, k-MIP attention emphasizes fully learnable attention patterns, single-stage training, and formal expressivity guarantees, whereas SpExphormer prioritizes aggressive memory reduction through two-stage training and sampled sparsification, enabling scalability to extremely large graphs, particularly when the input graph is already dense. Both methods highlight different trade-offs between computational efficiency, training complexity, and theoretical characterization.

## G.3    OTHER APPROACHES THAT EMPLOY LEARNABLE SPARSITY

Reformer (Kitaev et al., 2020) introduces an attention mechanism based on locality-sensitive hashing, in which keys and queries are first projected into a lower-dimensional space using a random matrix

and then hashed into angular buckets, after which attention is computed independently within each bucket. Similarly, Routing Transformer (Roy et al., 2021) clusters normalized key and query vectors at each self-attention step and restricts attention computation to tokens within the same cluster, where clusters are obtained via a spherical $k$-means algorithm with iteratively updated centroids. Clusterformer (Wang et al., 2021) follows an analogous strategy but applies Euclidean $k$-means to unnormalized key and query vectors. Sinkhorn Transformer (Tay et al., 2020) adopts a different approach by leveraging the Sinkhorn algorithm to approximate full attention, constructing a differentiable sorting network for each attention head that assigns relevant key blocks to query blocks before performing attention between the corresponding blocks. Clustered Attention (Vyas et al., 2020) employs a hierarchical strategy in which queries are clustered and attention is computed only for the cluster centroids with respect to all keys; each output token then inherits the attention result of its nearest centroid, reducing computational cost while preserving relevant information. Finally, we emphasize that our work was not inspired by prior uses of k-MIP attention in the sequence domain; instead, we develop this attention mechanism from the perspective of latent graph inference, where the goal is to discover an effective latent computational graph (Kazi et al., 2023; Borde et al., 2023b;a).

## H EXPERIMENTAL DETAILS

### H.1 DATASETS OVERVIEW

In this section, we provide an overview of the datasets used in our experiments. Table 4 summarizes the key characteristics of each dataset, including the number of graphs, average number of nodes and edges, whether the graphs are directed, the prediction level (graph, node, or link), the prediction task, and the evaluation metric used.

Table 4: Overview of the graph learning datasets used in this study.

| Dataset | # Graphs | Avg. # nodes | Avg. # edges | Directed | Prediction level | Prediction task | Metric |
|---|---|---|---|---|---|---|---|
| COCO-SP | 123,286 | 476.9 | 2,693.7 | No | ind. node | 81-class classif. | F1 score |
| PascalVOC-SP | 11,355 | 479.4 | 2,710.5 | No | ind. node | 21-class classif. | F1 score |
| Peptides-Func | 15,535 | 150.9 | 307.3 | No | graph | 10-task classif. | Avg. Prec. |
| Peptides-Struct | 15,535 | 150.9 | 307.3 | No | graph | 11-task regr. | MAE |
| Paris | 1 | 114k | 183k | No | transd. node | 10-class classif. | Accuracy |
| Shanghai | 1 | 184k | 263k | No | transd. node | 10-class classif. | Accuracy |
| LA | 1 | 241k | 343k | No | transd. node | 10-class classif. | Accuracy |
| London | 1 | 569k | 759k | No | transd. node | 10-class classif. | Accuracy |
| ShapeNet-Part | 16,881 | 2,616.2 | 20,929.6 | Yes | ind. node | 50-class classif. | F1 score |
| S3DIS | 23,585 | 4,096.0 | 131,072.0 | Yes | ind. node | 12-class classif. | mIoU |
| Cifar10 | 60,000 | 117.6 | 941.1 | Yes | graph | 10-class classif. | Accuracy |
| MalNet-Tiny | 5,000 | 1,410.3 | 2,859.9 | Yes | graph | 5-class classif. | Accuracy |

**COCO-SP** The COCO-SP dataset (CC BY 4.0 License) (Dwivedi et al., 2022) is part of the LRGB (Dwivedi et al., 2022). It is derived from the MS COCO image classification dataset (CC BY 4.0 License) (Lin et al., 2015) by constructing adjacency (hence planar) graphs on SLIC superpixels extracted from the images. The task is to classify each superpixel into one of 81 semantic segmentation classes. The dataset was downloaded from `https://www.dropbox.com/s/r6ihg1f4pmyjjy0/coco_superpixels_edge_wt_region_boundary.zip?dl=1` on 20 February 2025.

**PascalVOC-SP** The PascalVOC-SP dataset (Custom license for Pascal VOC 2011 respecting Flickr terms of use) (Dwivedi et al., 2022) is part of the LRGB (Dwivedi et al., 2022). It is derived from the Pascal VOC 2011 image classification dataset (Custom license for Pascal VOC 2011 respecting Flickr terms of use) (Everingham et al., 2010) by constructing adjacency (hence planar) graphs on SLIC superpixels extracted from the images. The task is to classify each superpixel into one of 21 semantic

segmentation classes. The dataset was downloaded from `https://www.dropbox.com/s/8x722ai272wqwl4/voc_superpixels_edge_wt_region_boundary.zip?dl=1` on 4 September 2024.

**Peptides-func** The Peptides-func dataset (CC BY-NC 4.0 License) (Dwivedi et al., 2022) is part of the LRGB (Dwivedi et al., 2022). It consists of atomic graphs of peptides obtained from the SATPdb dataset (CC BY-NC 4.0) (Singh et al., 2016). The task is multi-label graph classification, where each peptide is classified into one or more of 10 functional classes. The dataset was downloaded from `https://www.dropbox.com/s/ol2v01usvaxbsr8/peptide_multi_class_dataset.csv.gz?dl=1` on 20 February 2025.

**Peptides-Struct** The Peptides-Struct dataset (CC BY-NC 4.0 License) (Dwivedi et al., 2022) is part of the LRGB (Dwivedi et al., 2022). It consists of atomic graphs of peptides obtained from the SATPdb dataset (CC BY-NC 4.0 License) (Singh et al., 2016). The task is graph regression of 11 structural properties of the peptides. The dataset was downloaded from `https://www.dropbox.com/s/0d4aalmq4b4e2nh/peptide_structure_normalized_dataset.csv.gz?dl=1` on 20 February 2025.

**ShapeNet-Part** The ShapeNet-Part dataset (MIT License) (Yi et al., 2016) is a subset of the ShapeNet repository (ShapeNetCore v2 CAD models: non-commercial research license, see ShapeNet Terms of Use) (Chang et al., 2015), designed for 3D part segmentation tasks. It contains 16,881 3D shapes from 16 object categories. Each shape is annotated with 2-6 parts, with a total of 50 distinct part labels across all categories. The dataset is provided as 3D point clouds with corresponding part labels for each point. We transformed this point cloud segmentation task into a node classification task by constructing a directed k-NN graph on the point cloud where $k = 8$, which is the optimal $k$ found for ModelNet40 (Wu et al., 2015) in (Zheng et al., 2024). We use the train/test/validation split from PyTorch Geometric (Fey & Lenssen, 2019). The dataset was downloaded from PyTorch Geometric version 2.0.4.

**S3DIS** The Stanford Large-Scale 3D Indoor Spaces (S3DIS) dataset (CC BY 4.0) (Armeni et al., 2016) consists of the 3D point clouds of six large-scale indoor areas from three different buildings of Stanford university. The point features are the 3D positions and RGB values, and each point is labelled as one of 13 semantic classes. We transformed this point cloud segmentation task into a node classification task by constructing a directed k-NN graph on the point cloud where $k = 32$, which is the optimal $k$ found for this dataset in (Zheng et al., 2024). We use the train/test split from PyTorch Geometric (Fey & Lenssen, 2019), and randomly select 3,294 graphs from the training set for validation. This results in 16,997 training graphs, 3,294 validation graphs, and 3,294 test graphs. The dataset was downloaded from PyTorch Geometric version 2.0.4.

**Cifar10** The Cifar10 graph dataset (MIT License) (Dwivedi et al., 2023) was derived from the homonymous image classification dataset by constructing an 8-NN graph on SLIC superpixels extracted from the images. The node features are 5-dimensional and consist of 3 RGB values and (x,y)-coordinates for the superpixel. The edge features are 1-dimensional and consist of the node distance. The task is to classify the images into one of ten classes. The splits are the same as in the original dataset: 45K/5K/5K for training/validation/test respectively. The dataset was downloaded from PyTorch Geometric (Fey & Lenssen, 2019) version 2.0.4.

**MalNet-Tiny** MalNet-Tiny (CC-BY License) (Rampášek et al., 2022) is a subset of MalNet (CC-BY License) (Freitas et al., 2020) that is comprised of function call graphs derived from Android APKs. This subset contains 5,000 graphs with of up to 5000 nodes, each coming from benign software or 4 types of malware. The graphs are stripped of any original node or edge features, the task is to predict the type of the software based on the structure alone. (Rampášek et al., 2022). The dataset was downloaded from `http://malnet.cc.gatech.edu/graph-data/malnet-graphs-tiny.tar.gz` on 4 September 2024.

## H.2    Objective 1: Computational efficiency

This section provides detailed experimental methodology for the computational efficiency study presented in section 5.1. To comprehensively evaluate the computational efficiency of k-MIP attention, we designed a controlled experiment that systematically compares the runtime performance and memory consumption of different attention mechanisms across varying graph sizes.

Our experimental protocol involves sampling normally distributed key, query and value matrices of dimensions $N \times d_K$ for a range of node counts $N \in \{10^{i/2} \mid i = 4, \ldots, 12\}$, corresponding to graphs with sizes from $N = 100$ to $N = 10^6$ nodes. We fix the key dimension to $d_K = 10$.

We evaluate the computational efficiencies of the attention mechanisms under two distinct scenarios: (1) an *inference setting* where we perform only the forward pass without gradient tracking, simulating deployment scenarios where the model is used for prediction only, and (2) a *training setting* where we perform both forward and backward passes with full gradient tracking enabled, simulating the complete training pipeline including backpropagation through the attention mechanism.

For k-MIP attention, we consistently use $k = 10$ across all experiments.

All experiments are conducted on a single NVIDIA A100 40GB GPU using PyTorch 2.0 with CUDA 11.8. We measure wall-clock time for both forward and backward passes and peak GPU memory consumption. Each configuration is evaluated 5 times with different random seeds, and we report the mean $\pm$ standard deviation of both runtime and memory usage. The results are visualized in fig. 2, with the complete numerical data provided in Appendix J.1.

## H.3    LRGB

Tönshoff et al. (Tönshoff et al., 2023) have provided the most extensive hyperparameter search on the LRGB to date. Hence, to provide a fair comparison, we reproduced their numbers for GCN, GINE, GatedGCN, and GPS+Transformer, and employed their methodology for all models that were not included in their work.

Specifically, we conducted a "linear" hyperparameter search starting from a default configuration. For each hyperparameter, we evaluated its performance across a predefined range while keeping other parameters fixed. We then also evaluated the configuration obtained by combining the best-performing values for each hyperparameter. From all evaluated configurations, we selected the one with the highest validation performance as our final setting. For this hyperparameter search, we used a single run per configuration. For the final evaluation, we reported the averages and standard deviations across four different random seeds as specified by the LRGB (Dwivedi et al., 2022).

The hyperparameters and ranges were as detailed in table 5. For each configuration, the hidden dimension was chosen to be the maximum that made the model adhere to the official 500k parameter budget.

This differs from Tönshoff et al. (Tönshoff et al., 2023) in two ways. First, we don't search over the internal MPGNN and normalization for the graph transformer models. Instead, we always use GatedGCN as the internal MPGNN and LayerNorm as the normalization, as preliminary experiments showed that these configurations yield better performance. Second, we performed the full hyperparameter sweep and adhered to the official parameter budgeton the COCO dataset, rather than limiting it.

All other details are the same as in Tönshoff et al. (Tönshoff et al., 2023). The full configuration files are available in the provided code repository.

The data splits are the same as in Tönshoff et al. (Tönshoff et al., 2023) and Rampášek et al. (Rampášek et al., 2022). For Peptides-Func and Peptides-Struct, this means the official train/val/test splits are used. For COCO-SP and PascalVOC-SP, there are only train and val splits provided. We maintain the train split and divide the original val split into new val and new test split. The policy for this val/test splitting is as follows:

- Split total number of val graphs into 2 sets (val, test) with 50:50 using a stratified split proportionate to original distribution of data with respect to a meta label.

Table 5: Hyperparameter search ranges and default values for the LRGB experiment.

| Hyperparameter | Default | Range |
|---|---|---|
| Dropout | 0.1 | [0, 0.1, 0.2] |
| Depth | 8 | [6, 8, 10] |
| Learning rate | 0.001 | [0.001, 0.0005, 0.0001] |
| Head depth | 2 | [1, 2, 3] |
| Encoding | none | [none, LapPE, RWSE] |
| $k$ (only GPS + k-MIP) | 15 | [5, 15, 45] |
| Optimizer | AdamW | |
| Loss function | weighted CE | |
| # Epochs | 200 | |
| Batch size | 1 | |
| LR scheduler | cosine | |
| # Warmup epochs | 10 | |
| Weight decay | 0.0 | |
| Activation | GeLU | |
| MPNN layer | GatedGCN | |
| # Attention heads | 4 | |
| Hidden dimension | variable | |
| Attention dropout | 0.5 | |
| # Input FCLs | 0 | |
| # Output FCLs | 2 | |
| Graph pooling | mean | |
| Expander Degree (only Exphormer) | 3 | |
| # Virtual Nodes (only Exphormer) | 3 | |

- Each image is meta-labelled by majority voting of non-background ground truth node labels. Then new validation and new test sets are created with stratified sampling based on these meta-labels. This is done for preserving same distribution of node labels in both new val and new test.

Each experiment of the LRGB was conducted on a single 16GB V100 GPU. No run on Peptides-Func, Peptides-Struct, or PascalVOC-SP took longer than 4 hours to complete. On COCO, the training took roughly 8 hours for MPNNs, 40 hours for GPS+BigBird, and 20 hours for the other graph transformers.

## H.4 CITY-NETWORKS

On the City-Networks dataset, we tuned the hyperparameters on the Paris dataset and used those on all four datasets. For this hyperparameter search, we used the same methodology as for the LRGB experiment and ran all experiments ourselves. This means that, again, we conducted a "linear" hyperparameter search starting from a default configuration. For each hyperparameter, we evaluated its performance across a predefined range while keeping other parameters fixed. We then also evaluated the configuration obtained by combining the best-performing values for each hyperparameter. From all evaluated configurations, we selected the one with the highest validation performance as our final setting. For this hyperparameter search, we used a single run per configuration. For the final evaluation, we reported the averages and standard deviations across four different random seeds.

The hyperparameters and ranges are as detailed in table 6. For each configuration, the hidden dimension was chosen to be the maximum that made the model adhere to a 500k parameter budget.

We opted not to run the GPS+BigBird model on the LA dataset due to budget constraints. When processing the London dataset, memory constraints on our 80GB A100 GPU required us to reduce the model depth to 7 for GPS+k-MIP and 5 for GPS+Performer. While GPS+BigBird and Exphormer could have been run with a maximum depth of 3, we chose not to include these results as the reduced depth would have significantly compromised the fairness of the comparison.

Table 6: Hyperparameter search ranges and default values for the City-Networks experiment.

| Hyperparameter | Default | Range |
|---|---|---|
| Dropout | 0.1 | [0, 0.1, 0.2] |
| Depth | 8 | [6, 8, 12, 16] |
| Learning rate | 0.001 | [0.005, 0.001, 0.0005] |
| Head depth | 2 | [1, 2, 3] |
| Encoding | none | |
| $k$ (only GPS + k-MIP) | 15 | [5, 15, 45] |
| Optimizer | AdamW | |
| Loss function | weighted CE | |
| # Epochs | 1500 | |
| Batch size | 1 | |
| LR scheduler | cosine | |
| # Warmup epochs | 10 | |
| Weight decay | 0.0 | |
| Activation | GeLU | |
| MPNN layer | GatedGCN | |
| # Attention heads | 4 | |
| Hidden dimension | variable | |
| Attention dropout | 0.5 | |
| # Input FCLs | 0 | |
| # Output FCLs | 2 | |
| Graph pooling | mean | |
| Expander Degree (only Exphormer) | 3 | |
| # Virtual Nodes (only Exphormer) | 3 | |

Note in particular the larger range of depth values, which was chosen because the targets in the City-Networks dataset are constructed based on a 16-hop eccentricity (Liang et al., 2025), and they additionally observed that the performance of MPGNNs improves when increasing the depth to 16. The range of learning rates was also adjusted, because we found that a learning rate of 0.0001 was always too low on the LRGB.

Additionally, we enhanced the input features in two ways that improved the performance of the models. First, we normalized all input features to have zero mean and unit variance. Second, we concatenated sine and cosine encodings of each node's longitude and latitude to the input features. For a node with longitude $x$ and latitude $y$ (normalized over the training set to be in the range $[-0.9\pi, 0.9\pi]$) the encodings are defined as

$$
\boldsymbol{v}_{long} = \begin{pmatrix} \sin(x \cdot 2^0) \\ \sin(x \cdot 2^1) \\ \vdots \\ \sin(x \cdot 2^7) \\ \cos(x \cdot 2^0) \\ \cos(x \cdot 2^1) \\ \vdots \\ \cos(x \cdot 2^7) \end{pmatrix} \quad \text{and} \quad \boldsymbol{v}_{lat} = \begin{pmatrix} \sin(y \cdot 2^0) \\ \sin(y \cdot 2^1) \\ \vdots \\ \sin(y \cdot 2^7) \\ \cos(y \cdot 2^0) \\ \cos(y \cdot 2^1) \\ \vdots \\ \cos(y \cdot 2^7) \end{pmatrix} \tag{38}
$$

We used the train/val/test splits for each dataset provided by the authors (Liang et al., 2025).

The experiments on the Paris dataset were conducted on a single 40GB A100 GPU, whereas the experiments on the other datasets were conducted on a single 80GB A100 GPU. The runtimes for each model and dataset are provided in table 14.

Table 7: The hyperparameters used on ShapeNet-Part and S3DIS. GT stands for graph transformer.

| HP/Dataset | ShapeNet-Part | | S3DIS | |
| | MPNNs | GTs | MPNNs | GTs |
|---|---|---|---|---|
| **Metric** | F1 score | F1 score | mIoU | mIoU |
| **PE/SE type** | none | none | none | none |
| **Optimizer** | adamW | adamW | adamW | adamW |
| **Loss function** | weighted CE | weighted CE | weighted CE | weighted CE |
| **# Epochs** | 150 | 200 | 120 | 200 |
| **Batch size** | 8 | 8 | 8 | 8 |
| **LR** | 1e-3 | 5e-4 | 1e-3 | 5e-4 |
| **LR scheduler** | reduce_on_plateau | cosine | reduce_on_plateau | cosine |
| **Scheduler patience** | 10 | – | 10 | – |
| **Scheduler reduce factor** | 0.5 | – | 0.5 | – |
| **# Warmup epochs** | – | 10 | – | 10 |
| **Weight decay** | 1e-4 | 1e-4 | 1e-4 | 1e-4 |
| **Activation** | ReLU | ReLU | ReLU | ReLU |
| **# layers** | 5/10/15 | 8 | 5/10/15 | 8 |
| **MPNN layer** | – | GatedGCN | – | GatedGCN |
| **# Attention heads** | – | 4 | – | 4 |
| **Hidden dimension** | variable | variable | variable | variable |
| **Dropout** | 0.1 | 0.1 | 0.1 | 0.1 |
| **Attention dropout** | 0.0 | 0.0 | 0.0 | 0.0 |
| **# Input FCLs** | 0 | 0 | 0 | 0 |
| **# Output FCLs** | 2 | 2 | 2 | 2 |
| **Graph pooling** | sum | mean | sum | mean |
| **Expander Degree** | – | 3 | – | 3 |
| **# Virtual Nodes** | – | 3 | – | 3 |
| $k$ | – | 10 | – | 10 |

## H.5 POINT CLOUD DATASETS

**Dataset generation** Existing large-scale inductive graph benchmarks are often solvable with few-hop message passing neural networks, making them inadequate for evaluating attention mechanisms that benefit from long-range dependencies. To address this limitation and evaluate our method on challenging large-scale inductive tasks, we construct custom graph learning datasets from point cloud segmentation data.

We create graph learning tasks by adding k-NN graphs to two established point cloud segmentation datasets: ShapeNet-Part (Yi et al., 2016) and S3DIS (Armeni et al., 2016). This construction follows previous work that successfully applied graph neural networks to point cloud learning (Zheng et al., 2024; Liang et al., 2019; Shen et al., 2018; Wang et al., 2018). We set $k = 8$ for ShapeNet-Part and $k = 32$ for S3DIS, adopting the optimal values identified by Zheng et al. (Zheng et al., 2024).

This approach creates challenging inductive tasks that require capturing complex spatial relationships across entire point clouds, providing an ideal testbed for evaluating attention mechanisms in graph transformers.

**Hyperparameters** For the point cloud datasets, we did not have the resources to perform a full hyperparameter search. Instead, we chose the hyperparameters a priori based on the defaults used by Shirzad et al. (Shirzad et al., 2023), as detailed in table 7. For each configuration, we chose the hidden dimension to adhere to a parameter budget of 300k. Importantly, we used the same hyperparameters for all graph transformers. For the MPNNs, we ran versions with 5, 10, and 15 layers, and selected the best performing model based on the validation set.

For both ShapeNet-Part and S3DIS, we used the official train/val/test splits as provided by PyTorch Geometric (Fey & Lenssen, 2019).

The experiments on ShapeNet-Part were conducted on a single 16GB V100 GPU, except from Exphormer which was conducted on a single 40GB A100 GPU. The experiments took between 3 and 8 hours for the MPNNs, 8h for Exphormer and GPS+k-MIP, 12h for GPS+Performer, and 25h for GPS+BigBird.

The experiments on S3DIS were conducted on a single 40GB A100 GPU. They took roughly 30 hours for the MPNNs and 60h for the graph transformers.

### H.6 Computational infrastructure

The experiments for the LRGB and City-Networks benchmark were conducted on the infrastructure of an external cloud provider. The experiments for the point cloud datasets were conducted on an internal cluster.

A back-of-the-envelope calculation reveals that the experiments reported in the main text took approximately 3000 GPU hours to complete. Preliminary experiments, most notably those run on LRGB before we were aware of the critique of the experimental setting of Dwivedi et al. (Dwivedi et al., 2022) by Tönshoff et al. (Tönshoff et al., 2023), took approximately 1500 GPU hours to complete.

## I Additional results

### I.1 Not an approximation of full attention

While it is tempting to think of k-MIP attention as a way to approximate full attention, we examine in this section whether this is indeed the case.

**Experimental setup**  To study this question, we set up the following experiment. First, we train a GraphGPS model with k-MIP attention on the MalNet-Tiny dataset. Second, we extract the queries, keys, and values $\boldsymbol{Q}, \boldsymbol{K}, \boldsymbol{V}$ that are generated internally in the forward pass of the first 50 graphs with more than 1000 nodes in the test split of this dataset. We exclude the first layer and only consider queries, keys, and values from subsequent layers, as the first layer's inputs contain numerous duplicate values which would significantly distort our experimental results. Third, for these queries, keys and values, we compare the output of k-MIP attention $O_{\text{k-MIP}}^k = k\text{-MIP-Attn}(\boldsymbol{Q}, \boldsymbol{K}, \boldsymbol{V}) \in \mathbb{R}^{N \times d}$ for varying values of $k$ to the output of a full attention mechanism applied on the same matrices $O_{full} = \text{MHSA}(\boldsymbol{Q}, \boldsymbol{K}, \boldsymbol{V}) \in \mathbb{R}^{N \times d}$. For each value of $k$, we measure the average $\mathcal{L}_2$ distance between the corresponding rows of $O_{\text{k-MIP}}^k$ and $O_{full}$. Of this, we report the average and standard deviations over the 50 graphs, for each value of $k$.

**Results and Discussion**  The results of this experiment are shown in fig. 8. While it is tempting to think of k-MIP attention as a way to approximate full attention, the results demonstrate that k-MIP attention is a very poor approximation until $k$ is close to the number of nodes in the graph. For reference, for $k \leq 100$ the approximation was barely better than sampling from a multivariate normal distribution. Note that this does not contradict Theorem 2: even when every layer forms a poor approximation of the corresponding full attention layer, the composition of many such layers may still approximate the full-attention transformer.

**Inspecting the Attention Weights**  The reason for the poor approximation of the full attention mechanism can be found by inspecting the attention weights. In fig. 8b, we show the cumulative attention weights for the $k$ keys with the highest inner product for each query, for various values of $k$, as given by the expression

$$\sum_i^N \sum_{j \leq k} \text{softmax}_{1 \leq j \leq N}(\boldsymbol{q}_i^\top \boldsymbol{k}_{\sigma_i(j)}) \tag{39}$$

where each permutation $\sigma_i$ is chosen such that $\boldsymbol{q}_i^\top \boldsymbol{k}_{\sigma_i(1)} \geq \boldsymbol{q}_i^\top \boldsymbol{k}_{\sigma_i(2)} \geq \cdots \geq \boldsymbol{q}_i^\top \boldsymbol{k}_{\sigma_i(k)}$. Also, note that softmax normalization is over all $N$ keys here.

The results shows that, while the few highest-inner-product keys have a much larger attention weight than the other keys, they make up only a small fraction of the total attention weight. It takes values of

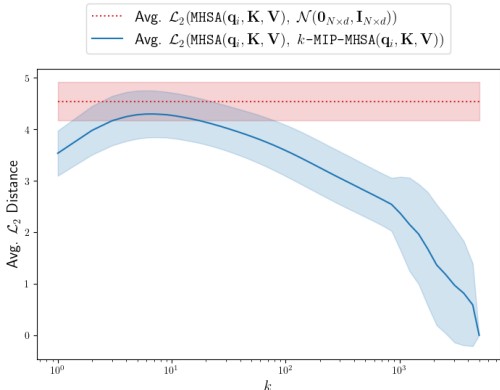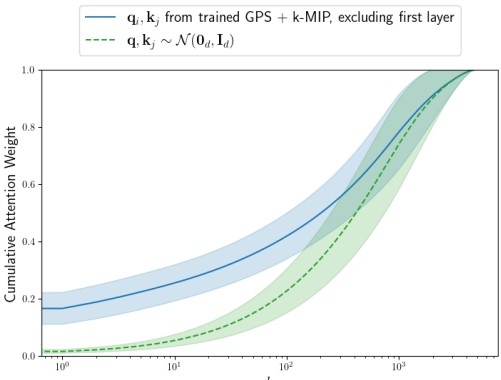

(a) The average $\mathcal{L}_2$ distance between the rows of `k-MIP-Attn`$(\boldsymbol{Q}, \boldsymbol{K}, \boldsymbol{V})$ and `MHSA`$(\boldsymbol{Q}, \boldsymbol{K}, \boldsymbol{V})$ for varying $k$, where the inputs are queries, keys, and values extracted from a trained GraphGPS model with k-MIP attention. The average and standard deviation are taken over the first 50 graphs with more than 1000 nodes in the test split of the MalNet-Tiny dataset. For comparison, when approximating the full attention output with samples from $\mathcal{N}(\boldsymbol{0}_{N \times d}, \boldsymbol{I}_{N \times d})$, the $\mathcal{L}_2$ distance was $4.5417 \pm 0.3756$.

(b) The cumulative attention weight for the $k$ keys with the highest inner product for each query, for varying $k$, where the inputs are queries and keys extracted from a trained GraphGPS model with k-MIP attention. The average and the standard deviation are taken over the first 50 graphs with more than 1000 nodes in the test split of the MalNet-Tiny dataset. For comparison, the cumulative attention weights are shown when the queries and keys are sampled from a multivariate normal distribution.

Figure 8: Analysis of k-MIP attention's relationship to full attention. **a.** The L2 distance between k-MIP attention and full attention outputs. **b.** The cumulative attention weight for the top-$k$ keys.

$k \approx 200$ to reach a cumulative attention weight of 0.5, and values of $k \approx 1000$ to reach a cumulative attention weight of 0.8. This implies that the many key-value pairs that do not belong to the $k$ keys with the highest inner product can still have a significant total impact on the output of the attention mechanism, which cannot be taken into account by k-MIP attention. This explains why the approximation is poor for values of $k$ that are not close to the number of nodes in the graph.

We conclude that k-MIP attention does not approximate the full attention mechanism. Instead, it should be thought of as a different way to perform attention, which has different properties and can be more efficient than full attention.

## I.2 INFLUENCE OF $k$

The hyperparameter $k$, which determines the number of keys that each query attends to, has the potential to significantly influence the performance and runtime of the model. In this section, we will therefore investigate the impact of $k$ on the runtime and prediction quality of a graph transformer with k-MIP attention.

One may be confident that the model's runtime will increase as $k$ grows, since the softmax operation in the attention mechanism must be computed over $k$ elements and $k$ weighted value vectors must be aggregated. Regarding performance, one may hypothesise that the model will improve with larger values of $k$, as it can take more context into account when processing each node. However, for larger values of $k$, there is also the risk of keys with low inner products introducing noise irrelevant to the query.

**Experimental setup** We train a GraphGPS model with k-MIP attention on four small-graph datasets with varying values of $k$. The other hyperparameters were chosen based on those found by Shirzad et al. (Shirzad et al., 2023). For each value of $k$, we conduct 5 runs and record the mean and standard deviation of both the test performance and the training epoch duration across these runs. All experiments are run with a single 16GB V100 GPU.

Table 8: Comparison of the performance of GraphGPS with k-MIP attention for varying $k$, on four small-graph datasets. Shown is the mean $\pm$ std over 5 runs. The **first** and second best results are highlighted.

| $k$ | Cifar10 Accuracy ↑ | MalNet-Tiny Accuracy ↑ | PascalVOC-SP F1 score ↑ | Peptides-Func AP ↑ |
|---|---|---|---|---|
| 1 | $74.37 \pm 0.56$ | $93.00 \pm 0.58$ | $35.57 \pm 1.06$ | $60.64 \pm 1.13$ |
| 2 | $74.84 \pm 0.41$ | $92.92 \pm 0.69$ | $\mathbf{37.65 \pm 0.70}$ | $62.97 \pm 0.27$ |
| 3 | $74.75 \pm 0.33$ | $92.88 \pm 0.41$ | $37.47 \pm 0.48$ | $63.13 \pm 0.65$ |
| 5 | $75.08 \pm 0.27$ | $\mathbf{93.54 \pm 0.52}$ | $37.30 \pm 0.99$ | $64.65 \pm 0.37$ |
| 10 | $75.05 \pm 0.32$ | $93.52 \pm 0.49$ | $36.58 \pm 1.50$ | $65.10 \pm 0.42$ |
| 20 | $75.12 \pm 0.31$ | $93.26 \pm 0.45$ | $35.80 \pm 1.60$ | $\mathbf{65.31 \pm 0.52}$ |
| 30 | $75.08 \pm 0.48$ | $92.62 \pm 0.70$ | $37.15 \pm 0.73$ | $65.14 \pm 0.76$ |
| 50 | $75.50 \pm 0.43$ | $92.78 \pm 0.75$ | $36.82 \pm 0.43$ | $64.90 \pm 0.84$ |
| 100 | $\mathbf{75.65 \pm 0.44}$ | $93.40 \pm 0.57$ | $36.90 \pm 0.33$ | $65.29 \pm 0.34$ |

Table 9: Comparison of the average training epoch duration in seconds of GraphGPS with k-MIP attention for varying $k$, on four small-graph datasets. Shown is the mean $\pm$ std over 5 runs. The **first** and second best results are highlighted.

| $k$ | Cifar10 Time (s) ↑ | MalNet-Tiny Time (s) ↑ | PascalVOC-SP Time (s) ↑ | Peptides-Func Time (s) ↑ |
|---|---|---|---|---|
| 1 | $\mathbf{135.34 \pm 8.72}$ | $\mathbf{22.09 \pm 1.19}$ | $16.14 \pm 0.95$ | $\mathbf{9.29 \pm 0.29}$ |
| 2 | $146.64 \pm 11.51$ | $23.96 \pm 0.89$ | $\mathbf{15.61 \pm 1.00}$ | $9.70 \pm 0.18$ |
| 3 | $146.50 \pm 20.20$ | $24.25 \pm 0.91$ | $16.81 \pm 1.11$ | $10.05 \pm 0.19$ |
| 5 | $147.94 \pm 14.23$ | $26.44 \pm 0.90$ | $17.12 \pm 0.86$ | $10.55 \pm 0.25$ |
| 10 | $146.60 \pm 19.97$ | $30.14 \pm 0.94$ | $17.70 \pm 0.89$ | $12.79 \pm 0.26$ |
| 20 | $142.29 \pm 20.11$ | $40.93 \pm 1.15$ | $22.58 \pm 0.62$ | $16.60 \pm 0.38$ |
| 30 | $149.70 \pm 19.76$ | $51.39 \pm 1.37$ | $26.92 \pm 1.35$ | $21.40 \pm 0.45$ |
| 50 | $166.97 \pm 13.62$ | $128.19 \pm 2.27$ | $73.42 \pm 2.05$ | $38.83 \pm 0.84$ |
| 100 | $241.80 \pm 19.67$ | $666.36 \pm 9.46$ | $318.91 \pm 5.11$ | $150.91 \pm 3.58$ |

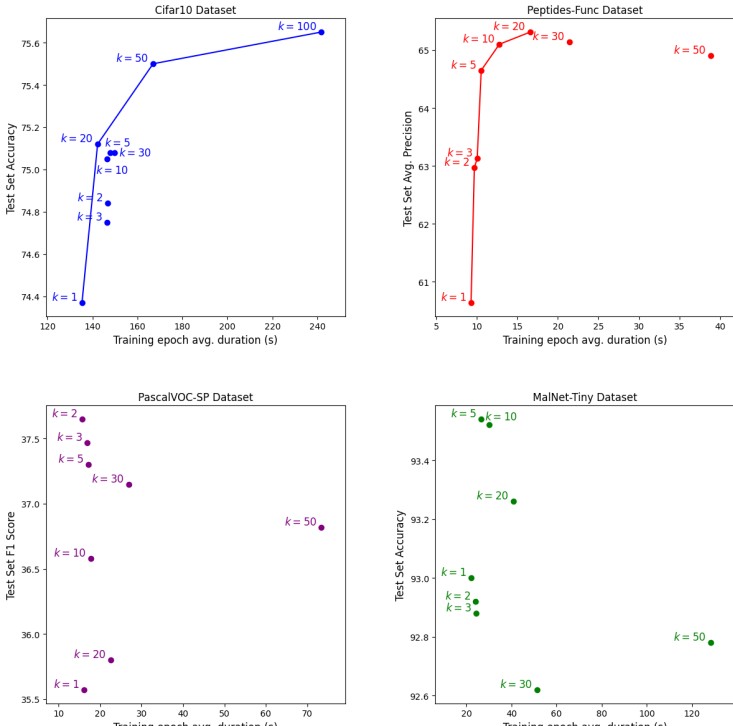

Figure 9: Scatter plots of the test performance against the average training epoch duration for varying values of $k$, for the datasets Cifar10, MalNet-Tiny, PascalVOC-SP, and Peptides-Func. The Pareto frontier is drawn for Cifar10 and Peptides-Func.

**Results**     The results of this experiment are presented in tables 8 and 9. It is clear that the runtime of the model increases with $k$. This increase is slow for small values of $k$, when the attention mechanism is only a small fraction of the total computation time, but becomes more pronounced for larger values

of $k$. The performance of the model, on the other hand, is relatively stable for varying values of $k$. Only for the Cifar10 and the Peptides-Func datasets is there a clear signal that very low values of $k$ are detrimental for the model's performance. For the other two datasets, the results are too noisy to draw a clear conclusion. For Cifar10 and Peptides-Func, this leads to $k$ presenting a trade-off between performance and efficiency, where a higher $k$ leads to better performance but also longer training times. This trade-off is visualised in fig. 9, where we drew the Pareto frontier for Cifar10 and Peptides-Func.

## J    DETAILED MEASUREMENTS

### J.1    OBJECTIVE 1: COMPUTATIONAL EFFICIENCY

Tables 10 to 12 present the exact data underlying the log-log plots shown in section 5.1. These tables provide the raw runtimes and memory usage measurements for each method and graph size.

Table 10: k-MIP attention with symbolic matrices: Detailed runtimes and peak GPU memory usage across different graph sizes, in the controlled experiment of section 5.1. All experiments were conducted on a single A100 GPU with 40GB memory. No gradient checkpointing was used. Averages are reported over 5 runs.

**k-MIP Attention with Symbolic Matrices**

|  | Inference Setting | Training Setting | | | |
| $N$ | Total runtime | Forward | Backward | Total | GPU memory (MB) |
|---|---|---|---|---|---|
| $10^{2.0}$ | 1.67 ms | 1.68 ms | 0.96 ms | 2.64 ms | 0.19 |
| $10^{2.5}$ | 1.69 ms | 1.79 ms | 0.98 ms | 2.76 ms | 0.58 |
| $10^{3.0}$ | 1.89 ms | 1.93 ms | 1.01 ms | 2.94 ms | 1.84 |
| $10^{3.5}$ | 2.14 ms | 2.23 ms | 1.04 ms | 3.27 ms | 5.80 |
| $10^{4.0}$ | 3.14 ms | 3.18 ms | 1.41 ms | 4.59 ms | 18.31 |
| $10^{4.5}$ | 6.38 ms | 6.62 ms | 1.84 ms | 8.46 ms | 57.90 |
| $10^{5.0}$ | 34.45 ms | 34.63 ms | 5.31 ms | 39.94 ms | 183.11 |
| $10^{5.5}$ | 0.231 s | 0.231 s | 0.0133 s | 0.245 s | 579.03 |
| $10^{6.0}$ | 2.146 s | 2.147 s | 0.0863 s | 2.233 s | 1831.06 |

Table 11: k-MIP attention without symbolic matrices: Detailed runtimes and peak GPU memory usage across different graph sizes, in the controlled experiment of section 5.1. All experiments were conducted on a single A100 GPU with 40GB memory. No gradient checkpointing was used. Averages are reported over 5 runs.

**k-MIP Attention: Naive (without Symbolic Matrices)**

|  | Inference Setting | Training Setting | | | |
| $N$ | Total runtime | Forward | Backward | Total | GPU memory (MB) |
|---|---|---|---|---|---|
| $10^{2.0}$ | 2.06 ms | 2.01 ms | 1.12 ms | 3.13 ms | 8.31 |
| $10^{2.5}$ | 2.00 ms | 2.00 ms | 1.00 ms | 3.00 ms | 8.71 |
| $10^{3.0}$ | 2.09 ms | 2.09 ms | 1.07 ms | 3.16 ms | 12.81 |
| $10^{3.5}$ | 2.19 ms | 2.20 ms | 1.12 ms | 3.33 ms | 49.00 |
| $10^{4.0}$ | 5.00 ms | 4.96 ms | 1.31 ms | 6.27 ms | 398.22 |
| $10^{4.5}$ | 30.97 ms | 31.76 ms | 1.72 ms | 33.48 ms | 3865.58 |
| $10^{5.0}$ | 0.262 s | 0.263 s | 0.004 s | 0.268 s | 36917.86 |
| $10^{5.5}$ | 2.622 s | 2.626 s | 0.009 s | 2.635 s | 36917.86 |
| $10^{6.0}$ | 26.670 s | 26.674 s | 0.086 s | 26.760 s | 36917.86 |

Table 12: Full attention: Detailed runtimes and peak GPU memory usage across different graph sizes, in the controlled experiment of section 5.1. All experiments were conducted on a single A100 GPU with 40GB memory. No gradient checkpointing was used. Averages are reported over 5 runs.

| | **Full Attention** | | | | |
| | Inference Setting | Training Setting | | | |
| $N$ | **Total runtime** | **Forward** | **Backward** | **Total** | **GPU memory (MB)** |
|---|---|---|---|---|---|
| $10^{2.0}$ | 0.64 ms | 0.75 ms | 1.04 ms | 1.79 ms | 16.42 |
| $10^{2.5}$ | 0.83 ms | 0.75 ms | 0.94 ms | 1.69 ms | 17.84 |
| $10^{3.0}$ | 0.67 ms | 0.76 ms | 1.04 ms | 1.80 ms | 31.70 |
| $10^{3.5}$ | 0.70 ms | 0.79 ms | 1.42 ms | 2.21 ms | 169.42 |
| $10^{4.0}$ | 2.61 ms | 2.69 ms | 7.41 ms | 10.10 ms | 1544.04 |
| $10^{4.5}$ | 24.76 ms | 24.36 ms | 47.10 ms | 71.46 ms | 15280.32 |
| $10^{5.0}$ | 0.246 s | OOM | OOM | OOM | OOM |
| $10^{5.5}$ | 2.500 s | OOM | OOM | OOM | OOM |
| $10^{6.0}$ | 24.496 s | OOM | OOM | OOM | OOM |

## J.2 CITY-NETWORKS

Tables 13 and 14 provide the accuracies and average runtimes for each model on the City-Networks dataset, respectively.

Table 13: Test accuracies of GPS + k-MIP and baselines on City-Networks (Liang et al., 2025). Shown is the mean ± std over four runs, except for the London dataset where we only ran one run for the graph transformer models. GPS + BigBird was not evaluated on LA due to long training times.

| | Paris | Shanghai | LA | London |
|---|---|---|---|---|
| # Nodes | 114k | 184k | 241k | 569k |
| # Edges | 183k | 263k | 343k | 759k |
| GCN | 52.93 ± 0.06 | 57.75 ± 0.24 | 56.65 ± 0.04 | 55.25 ± 0.06 |
| GINE | 53.36 ± 0.23 | 63.35 ± 0.20 | 58.21 ± 0.56 | 57.60 ± 0.20 |
| GAT | **55.83 ± 0.42** | **72.53 ± 0.23** | **65.53 ± 0.65** | 57.19 ± 1.09 |
| GatedGCN | 53.27 ± 0.10 | 68.80 ± 0.21 | 63.42 ± 0.12 | **61.47 ± 0.14** |
| GPS + BigBird | 53.53 ± 0.37 | 65.24 ± 0.17 | – | OOM |
| GPS + Performer | 54.06 ± 0.27 | 67.27 ± 0.17 | 61.64 ± 0.24 | 53.20 |
| GPS + Transformer | OOM | OOM | OOM | OOM |
| Exphormer | 51.40 ± 0.08 | 62.33 ± 0.16 | 58.15 ± 0.13 | OOM |
| GPS + k-MIP (ours) | 53.62 ± 0.22 | 66.94 ± 0.44 | 61.72 ± 0.35 | 56.05 |

Table 14: Training times of GPS + k-MIP and baselines on City-Networks (Liang et al., 2025). Shown is the mean over four runs, except for the London dataset where we only ran one run for the graph transformer models. All times were measured on a single 40GB A100 GPU for Paris, and a single 80GB A100 GPU for the other datasets.

| | Paris | Shanghai | LA | London |
|---|---|---|---|---|
| # Nodes | 114k | 184k | 241k | 569k |
| # Edges | 183k | 263k | 343k | 759k |
| | Time (h) | Time (h) | Time (h) | Time (h) |
| GCN | 0.051 | 0.066 | 0.089 | 0.193 |
| GINE | 0.048 | 0.065 | 0.087 | 0.183 |
| GAT | 0.071 | 0.093 | 0.125 | 0.278 |
| GatedGCN | 0.092 | 0.113 | 0.150 | 0.332 |
| GPS + BigBird | 11.50 | 19.21 | – | – |
| GPS + Performer | 2.62 | 5.08 | 7.87 | 13.66 |
| GPS + Transformer | – | – | – | – |
| Exphormer | 1.54 | 2.16 | 2.75 | – |
| GPS + k-MIP (ours) | 3.09 | 6.45 | 10.20 | 31.91 |

## K    COMPARISON WITH FLASHATTENTION

We acknowledge FlashAttention is a revelant baseline for scalable attention. To address this, we conducted preliminary experiments comparing k-MIP attention with FlashAttention, measuring runtime and peak memory usage under comparable settings on a single 40GB A100 GPU.

The inference setting consists of one forward pass without gradient tracking. Table 15 reports the measured runtime and peak memory. The training setting consists of one forward pass with gradient tracking and one backward pass. Table 16 reports the results.

Table 15: Inference runtime and memory comparison between k-MIP (fp32) and FlashAttention (fp16).

| $N$ | k-MIP (fp32) runtime | k-MIP (fp32) peak memory (MB) | FlashAttention (fp16) runtime | FlashAttention (fp16) peak memory |
|---|---|---|---|---|
| $10^{2.0}$ | 1.67 ms | 0.14 | 0.20 ms | 0.02 |
| $10^{2.5}$ | 1.69 ms | 0.45 | 0.17 ms | 0.13 |
| $10^{3.0}$ | 1.89 ms | 1.41 | 0.17 ms | 0.69 |
| $10^{3.5}$ | 2.14 ms | 4.47 | 0.18 ms | 2.17 |
| $10^{4.0}$ | 3.14 ms | 14.18 | 0.30 ms | 6.87 |
| $10^{4.5}$ | 6.38 ms | 44.63 | 1.80 ms | 6.87 |
| $10^{5.0}$ | 34.45 ms | 141.15 | 13.1 ms | 18.31 |
| $10^{5.5}$ | 0.231 s | 446.34 | 0.115 s | 57.91 |
| $10^{6.0}$ | 2.146 s | 1411.44 | 0.915 s | 183.11 |
| $10^{6.5}$ | 20.966 s | 4463.36 | 9.083 s | 579.03 |

Table 16: Training runtime and memory comparison between k-MIP (fp32) and FlashAttention (fp16).

| $N$ | k-MIP (fp32) total runtime | k-MIP (fp32) peak memory (MB) | FlashAttention (fp16) runtime | FlashAttention (fp16) peak memory |
|---|---|---|---|---|
| $10^{2.0}$ | 2.64 ms | 0.19 | 0.98 ms | 1.73 |
| $10^{2.5}$ | 2.76 ms | 0.58 | 1.01 ms | 5.17 |
| $10^{3.0}$ | 2.94 ms | 1.84 | 1.14 ms | 13.83 |
| $10^{3.5}$ | 3.27 ms | 5.80 | 1.29 ms | 43.23 |
| $10^{4.0}$ | 4.59 ms | 18.31 | 1.86 ms | 136.60 |
| $10^{4.5}$ | 8.46 ms | 57.90 | 7.89 ms | 428.88 |
| $10^{5.0}$ | 39.94 ms | 183.11 | 53.73 ms | 1352.44 |
| $10^{5.5}$ | 0.245 s | 579.03 | 0.374 s | 4273.56 |
| $10^{6.0}$ | 2.233 s | 1831.02 | 3.625 s | 13512.50 |
| $10^{6.5}$ | 21.232 s | 5790.30 | OOM | OOM |
| $10^{7.0}$ | 208.338 s | 18310.55 | OOM | OOM |

FlashAttention provides faster inference than k-MIP, achieving roughly a 2–2.5× speedup for smaller graphs. In training, k-MIP has an approximately 50% higher throughput for large graphs but is up to 2.7× slower for small graphs. However, we note that a direct comparison is not entirely fair due to several factors:

- **Tensor Core usage:** FlashAttention utilizes NVIDIA Tensor Cores, while KeOps (used by k-MIP) does not yet support these optimizations (Feydy et al., 2020). On the NVIDIA A100 GPU we benchmark on, Tensor Cores have an advertised peak throughput (in TFLOPS, for FP16) that is roughly 16× higher than the advertised peak throughput of the standard CUDA cores (FP32).

- **Precision:** FlashAttention uses FP16 while k-MIP uses FP32. Because FP16 typically offers about 2× higher FLOPS throughput than FP32, FlashAttention can achieve higher raw compute throughput.

- **Hardware optimization:** FlashAttention uses fused kernels, tiling, and optimized memory layouts, which are not yet implemented for k-MIP.

- **Framework maturity:** FlashAttention has undergone multiple rounds of optimization; k-MIP is a research prototype focused on validating the algorithmic idea.

We are currently working on engineering strategies for k-MIP, including FP16/INT8 quantization, Tensor Core support, fused and tiled kernels, and cache-aware memory layouts. Despite these differences, the comparison demonstrates that k-MIP attention is competitive, especially in terms of memory efficiency and scalability for large graphs.

