# OpenReview forum: "k-Maximum Inner Product Attention for Graph Transformers and the Expressive Power of GraphGPS"
_ICLR.cc/2026/Workshop/GRaM — ICLR 2026 Workshop GRaM Poster_

### Official Review · Reviewer_x66T · 2026-02-11
**Scalable but incrementally novel**

**Rating:** 6
**Confidence:** 3

**Review:**

**Summary:**
This paper introduces k-Maximum Inner Product (k-MIP) attention for graph transformers, where each query attends only to the k keys with the highest inner product scores. Using symbolic matrices, the method achieves linear memory complexity and roughly ten-fold speedup over full attention, enabling processing of graphs with 500k+ nodes on a single GPU.

**Strengths:**
1. Practical scalability to 500k+ node graphs.
2. The theoretical analysis is honest about its scope and limitations (e.g., Appendix D).
3. The insight that graph transformer expressiveness derives from encodings, not the attention mechanism, is well-argued and might be useful for the community.
4. Comprehensive experimental coverage across multiple benchmarks and thorough appendices.

**Weaknesses:**
1. Limited relevance to core GRaM themes.
2. Novelty is incremental: the attention mechanism and the symbolic matrix trick are not new, and the theoretical results are relatively direct extensions of prior work. Nevertheless, the author address such limitations.
3. The universal approximation theorem, has limited practical implications given the empirical findings that k-MIP doesn't approximate full attention layer-wise.

**General comments:**
1. The primary contribution is an efficient attention mechanism, which is more of a scalability contribution than a geometric one.  The paper does not  engage with symmetry preservation, equivariance, manifold-aware representations or other themes that I would expect in a GRaM paper. Yet, since this edition focus on "scale and simplicity". It might be suitable.

2. Novelty: the attention mechanism is not new ((Kreuzer et al., 2021) neither is the use of symbolic matrices to reduce memory cost, but I am not aware of previous works mixing both strategies.

3. Theory: Theorem proofs seems correct. The universal approximation result is acknowledged by the authors to follow almost directly from Yun et al. (2019), so the theory also has limited contribution. Yet,  the insight that expressiveness comes from encodings rather than the transformer architecture is valuable and well articulated.

4. Experiments: The authors follow Tönshoff et al.'s (2023) methodology for LRGB (I recommend citing the published version instead of the arvix one). Since the authors did not have the resources to use hyperparameter search on point cloud datasets, the comparisons on these benchmarks less reliable. On City-Networks, hyperparameters are tuned on Paris and transferred to all other cities, which is reasonable but may disadvantage methods differently across datasets. Most concerning is that the London result for graph transformers uses only a single run, limiting confidence in those numbers. Overall, the experiments are adequate for the workshop and demonstrate the advantage of using k-MIP.

5. Writing:  The paper is generally well-written and well-organized. The categorization of scalable graph transformers (Table 1) is helpful.

*Minor/typo:* equation 30 has a $\rho$ that was not introduced before. Should it be $\nu$?

**Pmlr Suitability:**

Yes

---

### Official Review · Reviewer_xSFM · 2026-02-21
**Interesting Idea with practical benefits, but limited evaluation and incremental/overclaimed theory**

**Rating:** 6
**Confidence:** 4

**Review:**

This paper introduces k-MIP attention, a sparse top-k inner-product attention mechanism for graph transformers. By leveraging symbolic matrix representations, the method achieves linear memory complexity while retaining quadratic compute. The authors integrate k-MIP into the GraphGPS framework, provide an expressive power analysis via an S-SEG-WL upper bound, and prove a universal approximation result for k-MIP transformers. Experiments are conducted on LRGB, City-Networks, and large-scale point cloud datasets. The work addresses an important scalability problem in graph transformers and is technically sound overall.

### Strengths:

* The practical contribution is clear and relevant: the memory-efficient implementation enables training on very large graphs on a single GPU.


* The method integrates cleanly into the GraphGPS framework and is well motivated from a design perspective.


* The empirical evaluation covers multiple benchmarks and demonstrates competitive performance.


### Weaknesses and concerns:

* The source of the reported runtime speedups is not fully explained. Since computational complexity remains quadratic, it would help to include analysis of memory traffic, FLOPs, or profiling details clarifying why symbolic matrices lead to large practical gains. At least some references to existing works would be nice here.


* The expressive power analysis appears somewhat incremental relative to prior SEG-WL-based analyses and existing work on positional encodings. The novelty relative to prior literature should be clarified more explicitly.


* There is a gap between theory and practice: the universal approximation theorem applies to pure k-MIP transformers, whereas experiments use the hybrid GraphGPS architecture, and the connection between the two is not fully discussed.


* Empirically, the method is competitive but not consistently superior to alternatives such as Performer or Exphormer. The main advantage seems to be scalability rather than improved predictive accuracy.


* Some evaluated datasets may not strongly require long-range information exchange, as suggested by strong MPNN baselines. Including more explicitly long-range tasks would strengthen the claims.


* The learning dynamics induced by the top-k selection are not analyzed. Since this operator restricts gradient flow, it would be useful to study learned sparsity patterns or compare against random sparse attention baselines.


* The relation to graph rewiring or latent edge-selection methods could be discussed more clearly.


### Minor comments:

* Figure 1 has low resolution.


* It is unclear why certain baselines (e.g., BigBird or Performer) are omitted from runtime comparisons in Figure 2.


* Some plots report training time only. Inference time could also be included.

**Pmlr Suitability:**

No

---

### Official Review · Reviewer_oZQL · 2026-02-23
**Theoretical Strengths, Underwhelming Efficiency Improvements**

**Rating:** 5
**Confidence:** 4

**Review:**

# Review

## Summary

This paper introduces k-MIP attention for graph transformers and integrates it into the GraphGPS framework. The authors provide a theoretical analysis of expressive power, showing (i) an upper bound via SEG-WL and (ii) a universal approximation result showing k-MIP transformers can approximate full-attention transformers. Empirically, they demonstrate improved memory scaling and practical speedups over full attention, enabling training on large graphs.

---

## Strengths

### 1. Strong theoretical framing

I particularly appreciate the careful theoretical analysis of expressive power. The connection between GraphGPS and SEG-WL provides useful clarification about where graph transformer expressivity originates.

The universal approximation result for k-MIP transformers is also well-formulated and clearly presented. Even if similar in spirit to prior universal approximation results, it strengthens the conceptual positioning of the method and shows that sparsification does not fundamentally reduce representational capacity.

Overall, the theoretical sections are thoughtful and contribute meaningfully to the literature on graph transformer expressivity.

---

## Weaknesses / Concerns

### 1. Limited asymptotic improvement in runtime

While the paper claims an “order-of-magnitude speedup,” the computational complexity remains quadratic in the number of nodes. The speedup shown in Figure 2 appears to be a constant-factor improvement rather than a reduction in asymptotic time complexity (the scaling slopes remain unchanged). Comparison to other scalable graph transformer methods is missing in Figure 2.

This is particularly important because there already exist graph transformer variants (e.g., Exphormer) that achieve O(Nd) runtime. In this context, improving the constant factor of an O(N^2) method is less compelling than achieving subquadratic complexity.

Given this landscape, the practical speedup over full attention feels underwhelming from an algorithmic standpoint.

---

### 2. Competitive but not superior performance

Empirically, k-MIP largely matches the performance of scalable O(Nd) methods such as Exphormer, GAT but does not clearly outperform them.

If a method retains quadratic computation while matching (but not exceeding) the performance of linear-time alternatives, it is unclear what the compelling advantage is beyond implementation simplicity.

In particular:

- On LRGB, Exphormer performs similarly or slightly better in some cases.
- On large-scale benchmarks, GAT consistently outperforms GPS + k-MIP both in accuracy and time.

This raises the question of whether the method offers a strong enough trade-off relative to existing sparse attention mechanisms.

---

### 3. Linear memory via known techniques

The linear memory footprint is an important practical contribution. However, this is largely enabled by symbolic matrix techniques (e.g., KeOps-style lazy evaluation), which are already established in the literature.

Thus, the memory improvement appears to stem more from an implementation strategy than from a fundamentally new algorithmic idea. While still useful, this somewhat reduces the conceptual novelty of the efficiency contribution.

---

## Questions for the Authors

1. Can the authors clarify in the abstract that runtime complexity remains quadratic? The current phrasing may suggest stronger asymptotic improvements than are actually achieved.
2. How does k-MIP compare in wall-clock time directly against GAT, Exphormer on the same hardware - addition to Figure 2?
3. Do the authors believe there are regimes (e.g., specific graph densities or feature dimensions) where k-MIP is preferable to existing O(Nd) attention mechanisms?

---

## Overall Assessment

This paper provides a strong theoretical treatment of expressivity. However, the efficiency contribution feels incremental in light of existing linear-time graph transformers. The method achieves linear memory and constant-factor runtime improvements over full attention but does not reduce asymptotic computational complexity and does not clearly outperform existing scalable alternatives.

**Recommendation: Borderline (weak reject / weak accept).**

The final decision depends on how the workshop values:

- Theoretical clarification of expressivity (a clear strength), versus
- The inferior computational efficiency compared to established scalable graph transformers, without a corresponding performance advantage.

**Pmlr Suitability:**

Yes

---

### Meta-Review · Area_Chair_rvJR · 2026-02-25

**Decision:**

Accept

**Metareview:**

This is a good submission that introduces k-MIP attention for Graph Transformers to boost scalability. The theory is solid and the linear memory advantages are great for large graphs. Reviewers liked the theoretical and empirical bits, but raised some concerns regarding novelty and runtime claims. The overall valuable contribution to GRaM and the soundness lead me to accept. I strongly recommend the authors to address/clarify the concerns in the final paper as good as possible.

**Relevance To Proceedings:**

Yes — suitable for PMLR (long paper)

**Relevance To Workshop:**

Yes — suitable for GRaM

---

### Decision · Program_Chairs · 2026-03-02

Accept (Poster)